# Malaria blood stage infection suppresses liver stage infection via host-induced interferons but not hepcidin

Hardik Patel [1], Nana K. Minkah[1,2], Sudhir Kumar [1], Gigliola Zanghi[1], Antonino Schepis[1], Debashree Goswami[1], Janna Armstrong[1], Biley A. Abatiyow[1], Will Betz[1], Laura Reynolds[1], Nelly Camargo[1], Amina A. Sheikh[1] & Stefan H. I. Kappe [1,2,3] ✉

Malaria-causing *Plasmodium* parasites first replicate as liver stages (LS), which then seed symptomatic blood stage (BS) infection. Emerging evidence suggests that these stages impact each other via perturbation of host responses, and this influences the outcome of natural infection. We sought to understand whether the parasite stage interplay would affect live-attenuated whole parasite vaccination, since the efficacy of whole parasite vaccines strongly correlates with their extend of development in the liver. We thus investigated the impact of BS infection on LS development of genetically attenuated and wildtype parasites in female rodent malaria models and observed that for both, LS infection suffered severe suppression during concurrent BS infection. Strikingly and in contrast to previously published studies, we find that the BS-induced iron-regulating hormone hepcidin is not mediating suppression of LS development. Instead, we demonstrate that BS-induced host interferons are the main mediators of LS developmental suppression. The type of interferon involved depended on the BS-causing parasite species. Our study provides important mechanistic insights into the BS-mediated suppression of LS development. This has direct implications for understanding the outcomes of live-attenuated *Plasmodium* parasite vaccination in malaria-endemic areas and might impact the epidemiology of natural malaria infection.

Malaria is a devastating disease caused by *Plasmodium* parasites, which resulted in more than 247 million clinical cases and >600 thousand deaths in 2021 alone[1]. *Plasmodium* parasites are transmitted as sporozoite stages by mosquito bite. Sporozoites infect hepatocytes and transform into liver stage trophozoites. These intracellular parasites then replicate and differentiate as asymptomatic liver stages (LS) to ultimately form the first generation of red blood cell-infectious merozoites, which initiate the symptomatic blood stage (BS) cycle of infection. Historically, the LS and BS of the *Plasmodium* life cycle have been studied independently[2]. However, accumulating evidence

suggests that host interactions of each stage can impact the other and influence the outcome of infection[2–4]. Indeed, a study conducted in rodent malaria models has demonstrated that concurrent BS infection inhibits LS development of a subsequent infection by modulating host responses[5,6]. Such interactions were predicted to lower the incidence of reinfection and to reduce the incidence of clinical malaria[5,7].

Inter-parasite stage interactions might not only impact the frequency and severity of natural infections but might also affect live-attenuated replication-competent whole LS parasite vaccine efficacy. In order to achieve high levels of protection against malaria infection,

[1]Center for Global Infectious Disease Research, Seattle Children's Research Institute, Seattle, WA, USA. [2]Department of Pediatrics, University of Washington, Seattle, WA, USA. [3]Department of Global Health, University of Washington, Seattle, WA, USA. ✉e-mail: stefan.kappe@seattlechildrens.org

these promising vaccines depend on substantial intra-hepatocytic development of the LS parasite immunogen[8,9]. Thus, efficacy of such vaccines could be affected by concurrent BS infection. Indeed, a recent clinical trial carried out using whole pre-erythrocytic parasite vaccination showed that the presence of BS infection completely abrogated vaccine efficacy[10]. Given the previous observations we therefore asked, using mouse models of malaria parasite infection, whether BS infection suppresses LS development and investigated the mechanism of LS suppression.

## Results

### Blood stage infection suppresses liver stage infection but not via hepcidin

We infected Balb/c mice with *P. yoelii* non-lethal XNL strain (*Py*) BS parasites (Fig. 1A). Four days later, the *Py* BS-infected mice alongside naive control mice were infected with sporozoites of a genetically attenuated, replication-competent *Py* parasite strain expressing luciferase (*Py*GAP[luc]). This parasite strain undergoes near complete LS development and is only distinguishable from wild-type parasites by a lack of exo-erythrocytic merozoite differentiation[11]. Liver stage infection was then measured at 24 and 43 h post sporozoite infection (hpi), using quantitation of LS bioluminescence in the liver (Fig. 1A). We observed an exponential increase in the LS bioluminescence signal from 24 to 43 hpi in BS-naïve control mice (GAP LS[luc]), indicating robust LS development. However, LS infection was severely suppressed by concurrent *Py* BS infection (*Py* BS + GAP LS[luc], Fig. 1A). These data show that suppression of *Py* GAP LS infection occurs in the presence of *Py* BS infection. Thus, we next asked what host factors could mediate this effect.

It has been previously shown that *Plasmodium berghei* BS (*Pb* BS) infection significantly elevates the levels of the iron regulatory hormone hepcidin[5,12] and that elevated hepcidin levels inhibit LS infection[5]. The major pathways that drive hepcidin expression are interleukin-6 (IL-6) and bone morphogenetic protein (BMP) receptor-mediated signaling during inflammation[13,14]. Thus, we first determined whether *Py* BS infection leads to an increase of hepcidin, similar to *Pb* BS, and whether blocking these two pathways prevent the elevation of hepcidin. We treated *Py* BS-infected Balb/c mice individually with anti-IL-6 antibody and LDN193189 (a well-studied BMP type-I receptor inhibitor[13]) or with a combination of both and measured circulating hepcidin levels in the blood by competitive binding ELISA in comparison to a control treatment (Fig. 1B). We observed that *Py* BS-infected mice showed indeed significant elevation of circulating hepcidin levels from baseline and that blocking individual signaling pathways substantially dampened the elevation of hepcidin levels (Fig. 1B). Importantly, the simultaneous blockade of both pathways maintained hepcidin levels at the baseline that we observed in uninfected control mice. BS parasitemia remained unaffected by these treatments (Fig. 1C). We next analyzed whether BS infection-mediated elevation of hepcidin levels are indeed the underlying cause for suppression of LS infection. To this end, we used the anti-IL-6/LDN193189 treatment in *Py* BS-infected mice followed by infection with *Py*GAP[luc] sporozoites (Fig. 1D). Strikingly, we did not observe rescue of GAP LS infection (Fig. 1E), despite maintaining hepcidin expression levels in the liver at a baseline similar to mice without *Py* BS infection (Fig. 1F).

The LS-suppressing role of hepcidin might be parasite and/or mouse strain-specific[5]. Therefore, we conducted our studies utilizing the experimental conditions reported previously[5]. C57BL/6 mice infected with *Pb* NK65 BS parasites also showed elevated expression of hepcidin and treatment with anti-IL-6/LDN193189 maintained hepcidin levels at the baseline both in the liver and blood without affecting BS parasitemia (Fig. 2A–C). Yet, the treatment did not restore *Pb* ANKA[luc] wild-type LS infection during concurrent BS infection (Fig. 2D, E), although it maintained hepcidin levels at a similar baseline observed without BS infection (Fig. 2F). Our results therefore demonstrate that

*Plasmodium* BS infection-induced hepcidin does not suppress LS infection, irrespective of the parasite species used for BS and LS infection or the mouse strains which received infection[5,7].

### *P. yoelii* BS infection suppresses LS infection via IFNγ

We next set out to identify the causative factors which mediate suppression of LS infection during a concurrent BS infection. It is known that BS malaria induces inflammatory responses, including interferon (IFN) responses[15–17]. Furthermore, we and others have previously shown that type I interferon (IFN-I) signaling and IFNγ inhibit LS infection[18–20]. Therefore, we next examined the role of *Py* BS infection-induced IFN-I signaling or IFNγ in inhibiting LS growth of *Py* GAP. We used IFN-I receptor (IFNAR1)-blocking antibodies or anti-IFNγ-neutralizing antibodies in the above-described *Py* BS/*Py* GAP LS co-infection model. Interestingly, while blocking of IFNAR1 signaling did not restore LS infection (Fig. S1), neutralizing IFNγ significantly reversed the suppression of *Py* GAP LS infection during concurrent *Py* BS infection (Fig. 3A, B). The BS parasitemia remained unaffected by treatments (Fig. 3C)[17]. In order to determine whether the effect of BS-induced IFNγ is mouse strain-specific, we also performed the experiments in C57BL/6 mice and found that the *Py* BS infection-induced IFNγ response also suppressed *Py* GAP LS infection. This finding was further substantiated in IFNγ[-/-] knockout C57BL/6 mice (Fig. 3D), which showed little suppression of LS infection during concurrent BS infection.

We next asked whether the BS-induced IFNγ would suppress LS infection of wild-type *Plasmodium* parasites. Balb/c mice with ongoing *Py* BS infection were infected with wild-type *Py*[GFP-luc] sporozoites (Fig. 3A). Mice with concurrent BS infection showed severe suppression of LS infection and neutralizing IFNγ significantly restored *Py* LS development (Fig. 3E). Microscopic examination of LS-infected liver sections between 43–46 hpi further confirmed that neutralizing IFNγ restores normal development of *Py* LS in the presence of *Py* BS infection (Fig. S2).

### IFN-γ does not affect early hepatocyte infection but suppresses LS development

We next investigated at what stage of LS infection IFNγ exerts its suppressive effect. We selected two readout time points: Six hpi, which encompasses sporozoite invasion of hepatocytes and transformation of intracellular sporozoites into early LS trophozoites and 48 hpi, which measures near full LS development. We utilized in vitro *Py* hepatoma cell line infections[21], where HepG2-CD81 cells were treated with human recombinant IFNγ at 16 h, or 3 h pre-infection or during the sporozoite addition to HepG2-CD81 cells for up to 3 hpi (Fig. 4A). Interestingly, we did not observe any difference in the number of infected cells between control and IFNγ-treated cells when measuring HepG2-CD81 infection at 6 hpi for all treatment timeframes (Fig. 4B, C). However, we did observe ~50% reduction in the number of LS-infected cells in all treated groups at 48 hpi compared to the control (Fig. 4D, E). Moreover, the size of the remaining LS was also significantly reduced in IFNγ-treated, infected cells when compared to the control (Fig. 4F, G). These data suggest that IFNγ does not affect the sporozoite invasion process or early events in the establishment of LS infection but substantially eliminates later LS forms and suppresses the growth of the remaining LS.

### *P. berghei* BS infection suppresses LS development via both type I and II interferons

We next sought to determine whether the BS-induced, IFNγ-mediated suppression of LS development is a common mechanism of action across different *Plasmodium* species. C57BL/6 mice were infected with *Pb* NK65 BS parasites and treated with IFNγ-neutralizing antibody or isotype control. Four days later these mice alongside naïve control mice were infected with wild-type *Pb* ANKA[luc] sporozoites (Fig. 5A). We

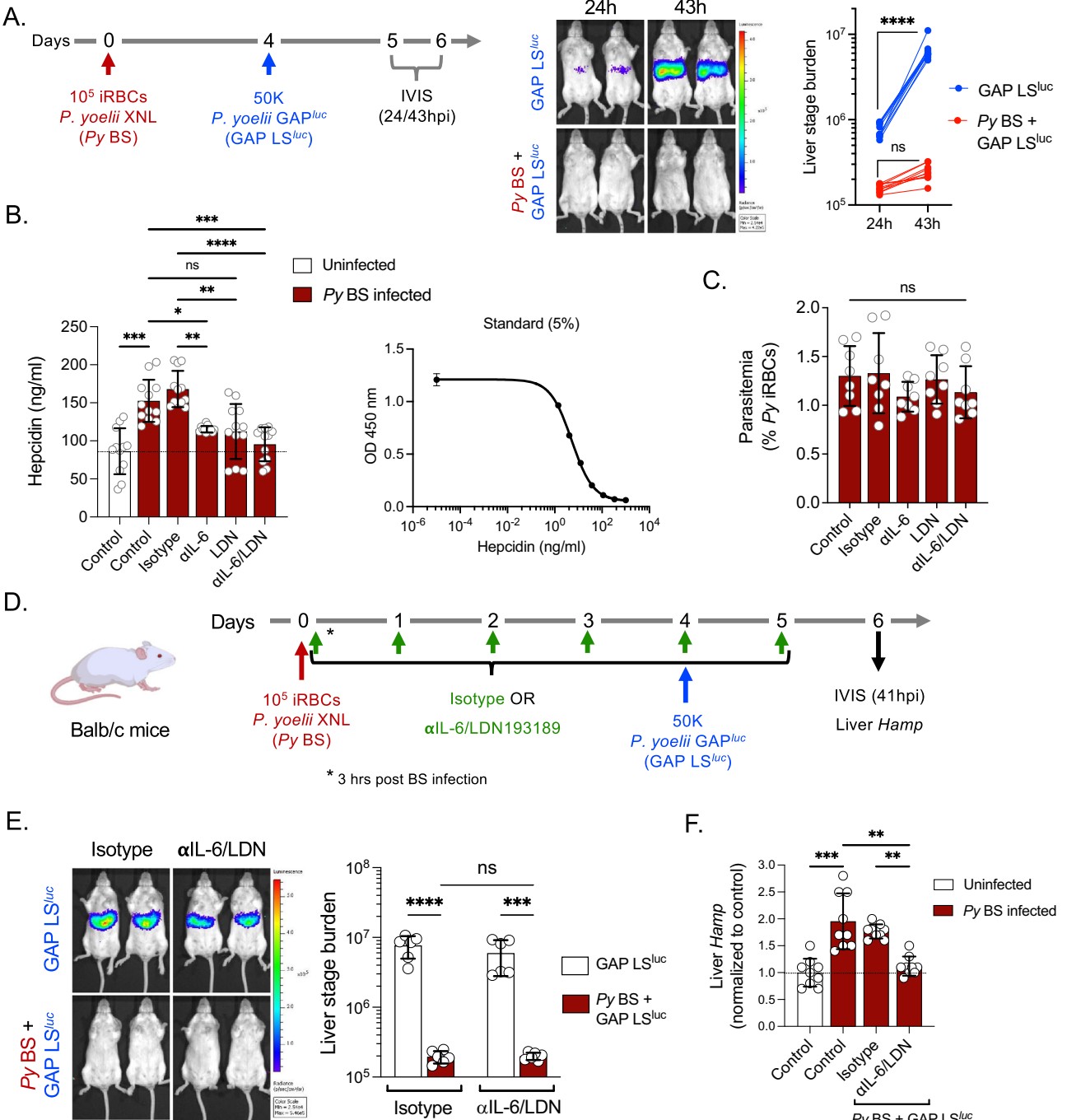

**Fig. 1 | *P. yoelii* blood stage infection-induced hepcidin does not suppress *P. yoelii* GAP liver stage development. A** Balb/c mice were infected with 10⁵ *Py* XNL infected RBCs (iRBCs) and 4 days later were infected with 50,000 luciferase-expressing *Py* GAP^luc sporozoites (*Py* BS + GAP LS^luc). Control mice were infected with 50,000 *Py* GAP^luc sporozoites without giving prior BS infection (GAP LS^luc). Parasite liver stage burden was measured through bioluminescence by IVIS at 24 and 43 h post sporozoite infection (hpi) and represented as total flux (p/s). *n* = 10 mice per group. ****P < 0.0001. **B** Balb/c mice were infected with 10⁵ *Py* XNL iRBCs and were treated with isotype, anti-IL-6 monoclonal antibodies, LDN193189 inhibitor, or anti-IL-6/LDN193189 inhibitor in combination every day starting from day 0 (3 h after inoculation with iRBCs) until day 4. Blood was collected on day 4 and circulating hepcidin was measured by ELISA in uninfected and *Py* BS-infected Balb/c mice. *n* = 12 mice per group. *P = 0.04, **P = 0.003, ***P < 0.001, ****P < 0.0001.

**C** The parasitemia was determined by counting of *Py* infected RBCs (% *Py* iRBCs) in Giemsa-stained thin blood smears. *n* = 8 mice per group. **D** Experimental layout. Control (GAP LS^luc) and *Py* BS-infected (*Py* BS + GAP LS^luc) mice were treated with isotype or anti-IL-6/LDN193189 inhibitor and were infected with 50,000 *Py* GAP^luc sporozoites on day 4 post BS infection. **E** Parasite stage burden by IVIS (*n* = 6 mice per group) ***P = 0.0006, ****P < 0.0001. **F** relative hepcidin expression (*Hamp*) by qPCR were measured at 41 h after sporozoite infection. *n* = 9 mice per group. **P = 0.004, ***P = 0.0006. **A, E** Two-way ANOVA with Tukey's multiple comparison test for comparing groups with two variables. **B, C, F** Kruskal–Wallis test followed by Dunn's multiple comparison test. Results are combined and represented as means ± SD from three (**A, B, F**) or two (**C, E**) independent experiments. Source data are provided as a Source data file. The graphical illustration of the mouse in (**D**) was made using BioRender.com.

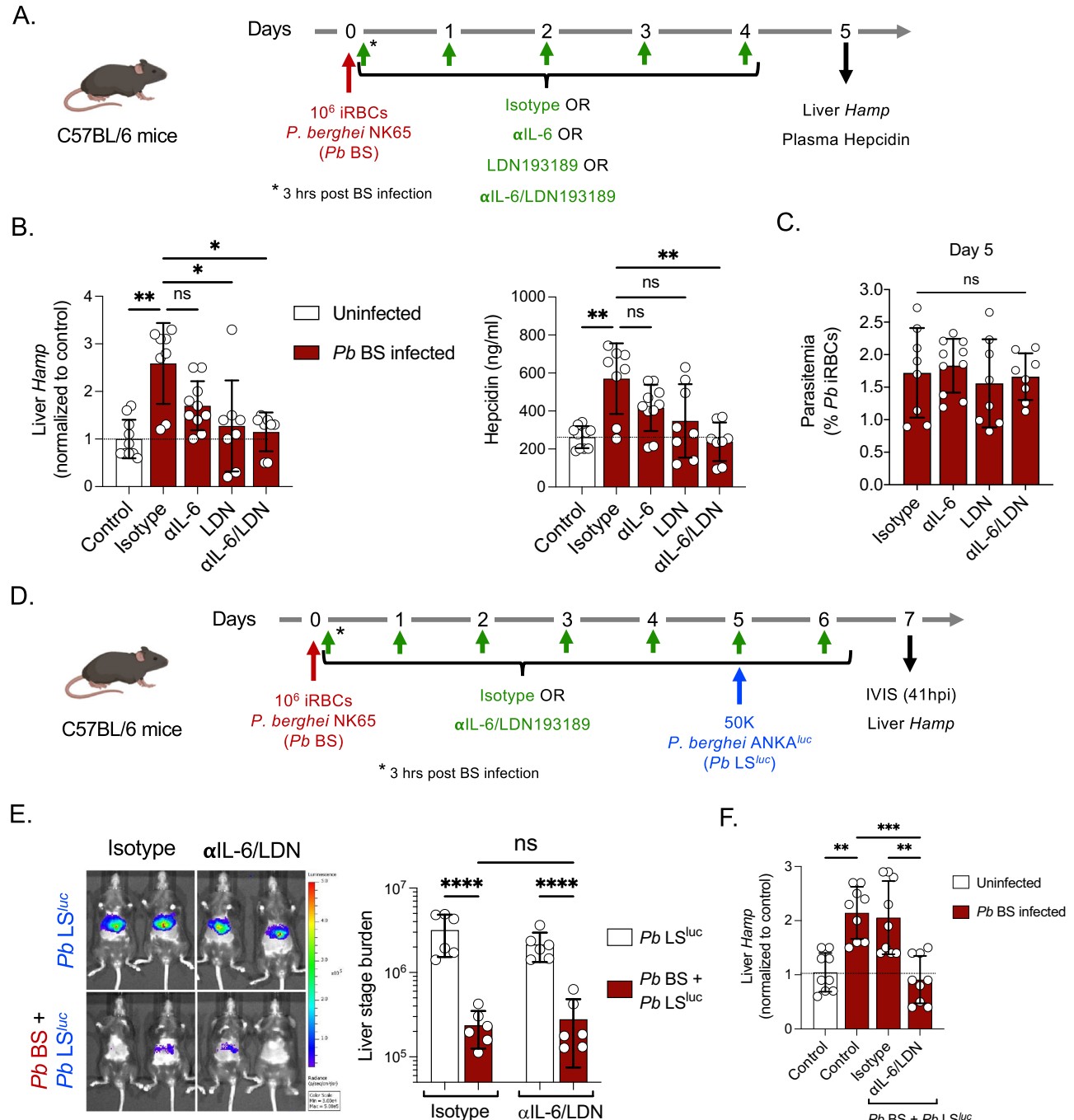

**Fig. 2 | *P. berghei* blood stage infection-induced hepcidin does not suppress *P. berghei* liver stage development.** **A** Experimental layout. C57BL/6 mice were infected with $10^6$ *Pb* NK65 iRBCs and treated with isotype, anti-IL-6 monoclonal antibodies, LDN193189 inhibitor, or anti-IL-6/LDN193189 inhibitor in combination every day starting from day 0 (3 h after inoculation with infected RBCs) until day 4. The liver and blood were collected on day 5. **B** The *Hamp* gene expression in the liver (left panel) and circulating hepcidin in the blood (right panel) were measured by qPCR and ELISA, respectively. *n* = 8–10 mice per group. \**P* < 0.05, \*\**P* < 0.01. **C** Blood stage parasitemia was determined by counting of *Pb* infected RBCs (% *Pb* iRBCs) in Giemsa-stained thin blood smears. *n* = 8–10 mice per group. **D** Experimental layout. Control (*Pb* LS^luc^) and *Pb* BS-infected (*Pb* BS + *Pb* LS^luc^) mice

were treated with isotype or anti-IL-6/LDN193189 inhibitor and were infected with 50,000 *Pb* ANKA^luc^ sporozoites on day 5 post BS infection. **E** Parasite stage burden by IVIS (*n* = 6 mice per group). \*\*\*\**P* < 0.0001. **F** Relative hepcidin expression (*Hamp*) by qPCR were measured at 41 h after sporozoite injection. *n* = 9 mice per group. \*\**P* < 0.01, \*\*\**P* = 0.0007. **B**, **C**, **F** Kruskal–Wallis test followed by Dunn's multiple comparison test. **E** Two-way ANOVA with Tukey's multiple comparison test for comparing groups with two variables. Results are combined and represented as means ± SD from three (**B**, **F**) or two (**C**, **E**) independent experiments. Source data are provided as a Source data file. The graphical illustration of the mouse in (**A**) and (**D**) was made using BioRender.com.

did not observe restoration of *Pb* LS development with IFNγ neutralization in the *Pb* BS + *Pb* LS group (Fig. 5B). These data indicated that *Pb* BS-induced suppression of *Pb* LS might not be solely or not at all mediated by IFNγ, contrary to what we observed in *Py* BS/*Py* LS co-

infection (Fig. 3). To determine whether the outcome is dependent on the parasite species used for BS infection, groups of mice were infected with either *Py* or *Pb* BS parasites and treated with IFNγ-neutralizing antibody. Four days later, both groups and BS-naïve control mice were

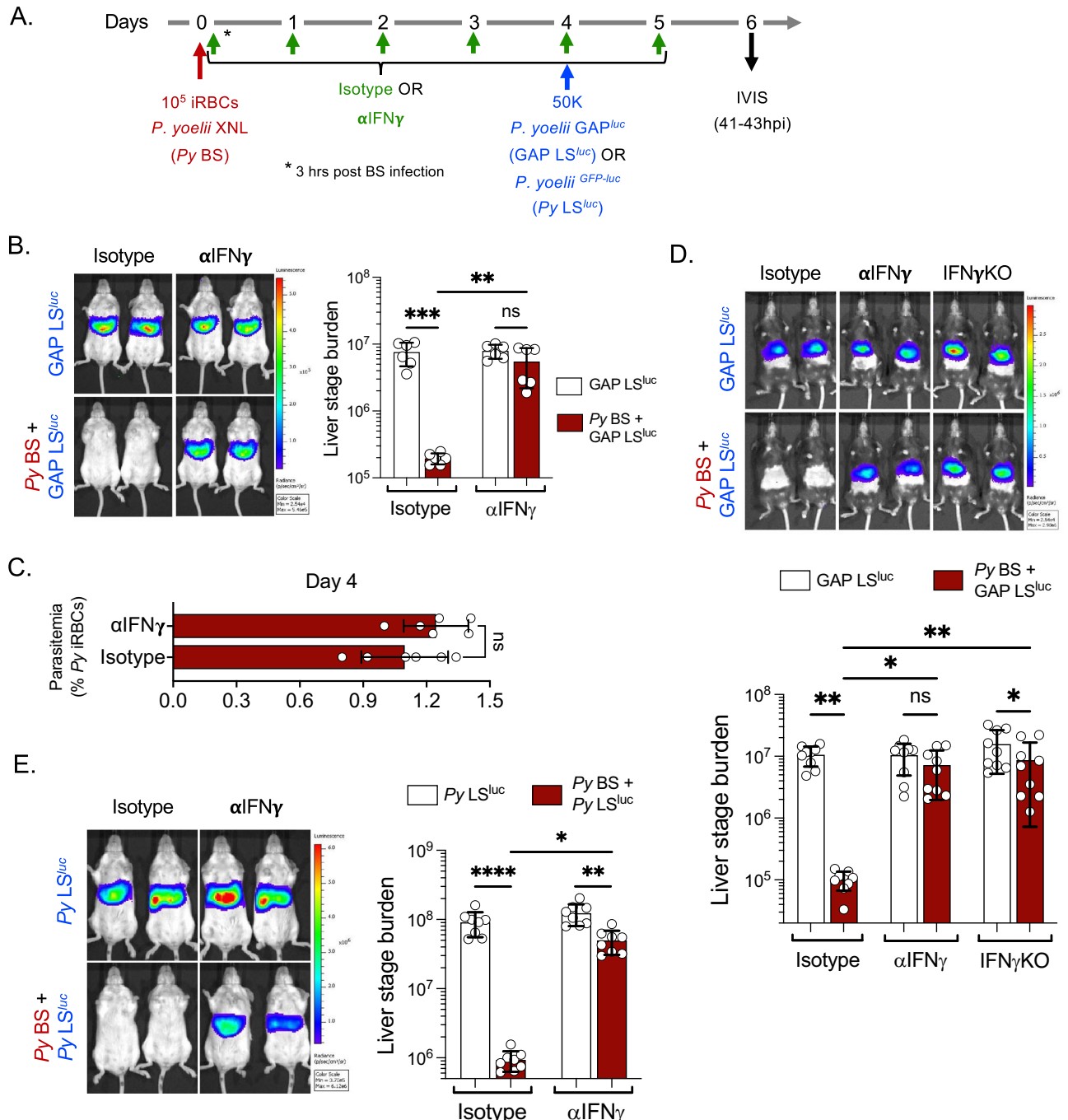

**Fig. 3 | *P. yoelii* blood stage infection-induced IFNγ suppresses liver stage development. A** Experimental layout. Control (LS^luc) and *Py* BS-infected (*Py* BS + LS^luc) mice were treated with Isotype or anti-IFNγ antibodies as shown in the scheme and were infected with 50,000 *Py* GAP^luc or wild-type *Py*^GFP-luc sporozoites on day 4 post BS infection. **B** Parasite liver stage burden measured in Balb/c mice by IVIS at 41 h after *Py* GAP^luc sporozoite infection and represented as total flux (p/s). *n* = 6 mice per group. **P = 0.005, ***P = 0.0002. **C** Blood stage parasitemia measured in Balb/c mice on day 4 post *Py* BS infection. *n* = 6 mice per group. **D** Parasite liver stage burden measured in C57BL/6 and IFNγ^-/- mice by IVIS at 41 h after *Py* GAP^luc sporozoite infection and represented as total flux (p/s). *n* = 9 mice per group. *P < 0.05, **P < 0.01. **E** Parasite liver stage burden in Balb/c mice was measured by IVIS at 43 h after wild-type *Py*^GFP-luc sporozoites infection. *n* = 6 mice per group. *P = 0.01, **P = 0.002, ****P < 0.0001. **B, D, E** Two-way ANOVA with Tukey's multiple comparison test for comparing groups with two variables. **C** Two-sided non-parametric Mann–Whitney U-test. Results are combined and represented as means ± SD from two (**B, C, E**) or three (**D**) independent experiments. Source data are provided as a Source data file.

infected with *Pb* ANKA^luc sporozoites and LS infection was measured at 41 hpi. We found that while in the *Py* BS + *Pb* LS^luc group of mice, *Pb* LS development was significantly restored upon neutralizing IFNγ, the same did not restore LS development in the *Pb* BS + *Pb* LS^luc group (Fig. 5C). These results indicated that *Py* BS and *Pb* BS infections might provoke distinct host responses which result in suppression of LS

development and also that both *Py* LS and *Pb* LS are susceptible to IFNγ-mediated suppression.

We therefore analyzed host cytokine responses in C57BL/6 mice infected with either *Py* or *Pb* BS parasites. Blood was collected at different time points during the course of infection and subjected to a comparative multi-cytokine kinetic analysis (Fig. 5D). We focused on

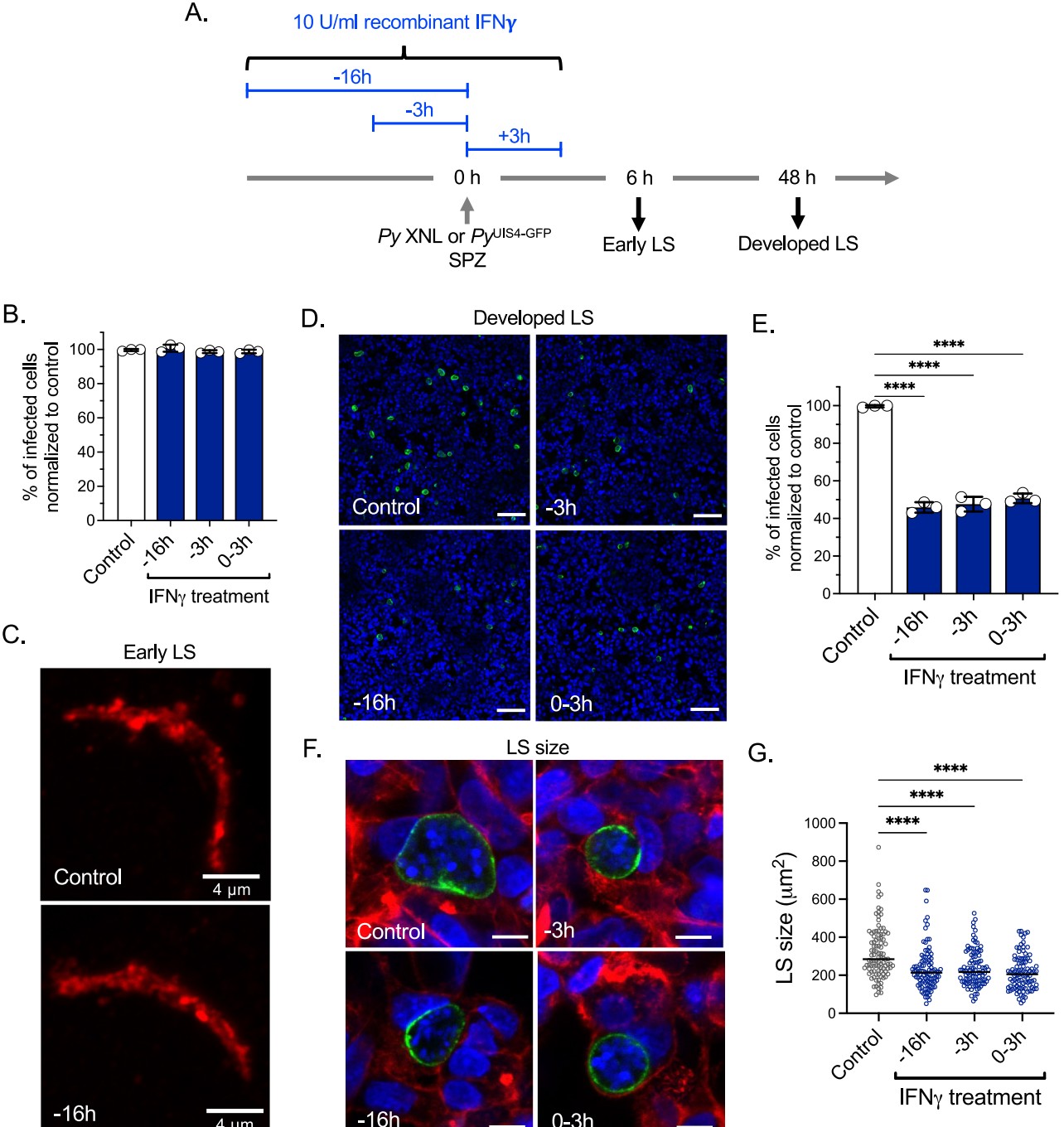

**Fig. 4 | IFNγ does not affect early phases of hepatocyte infection but suppresses LS development. A** Experimental layout. HepG2-CD81 hepatoma cells were treated with 10 U/ml human recombinant IFNγ at 16 h, or 3 h pre-infection or during *Py* or *Py*^UIS4-GFP sporozoite addition to the cells for up to 3 h post infection (hpi). The suppressive effect of IFNγ was determined at 6 hpi (early LS), which encompasses sporozoite invasion of hepatocytes and transformation of intracellular sporozoites into LS trophozoites and 48 hpi, which measures near full LS development. **B, C** The number of infected cells between control and IFNγ treated groups at 6 hpi (early LS) determined by UIS4 staining. Scale bar, 4 μm. *n* = 3 independent experiments per group. **D, E** Quantification of UIS4-GFP positive infected cells between control and IFNγ treated groups at 48 hpi (developed LS) determined by live imaging after staining with Hoechst 33342. Scale bar, 100 μm. *n* = 3 independent experiments per group. ****P < 0.0001. **F, G** Quantification of LS size at 48 hpi by measuring area of UIS4-GFP after staining with Phalloidin rhodamine and Hoechst 33342. Scale bar, 10 μm. *n* = 100 *Py* LS count per group. ****P < 0.0001. **B, E, G** One-way ANOVA with Tukey's multiple comparison test. Results are combined and represented as means ± SD from three independent experiments (**B**–**E**) or represented from one of the three independent experiments with similar results obtained (**F, G**). Source data are provided as a Source data file.

cytokines that were previously shown to inhibit parasite growth, are involved in causing liver injury or have immunomodulatory roles[18,19,22–28], namely IFNγ, IL-6, IL-12p40, IFNα, TNFα, IL-1α, IL-10, IFNβ, IL-1β, IL-12p70, and TGFβ. We did not observe significant expression of IFNβ, IL-1β, IL-12p70, and TGFβ during the course of both *Py* BS and *Pb*

BS infection (data provided in the Fig. 5 tab of Source data file). In contrast, IFNγ, IL-6, IL-12p40, IFNα, TNFα, IL-1α, and IL-10 were markedly upregulated from their baseline expression during both or either one of the parasite BS infection (Fig. 5D), with variation in the expression kinetics of these cytokines. We observed high levels of

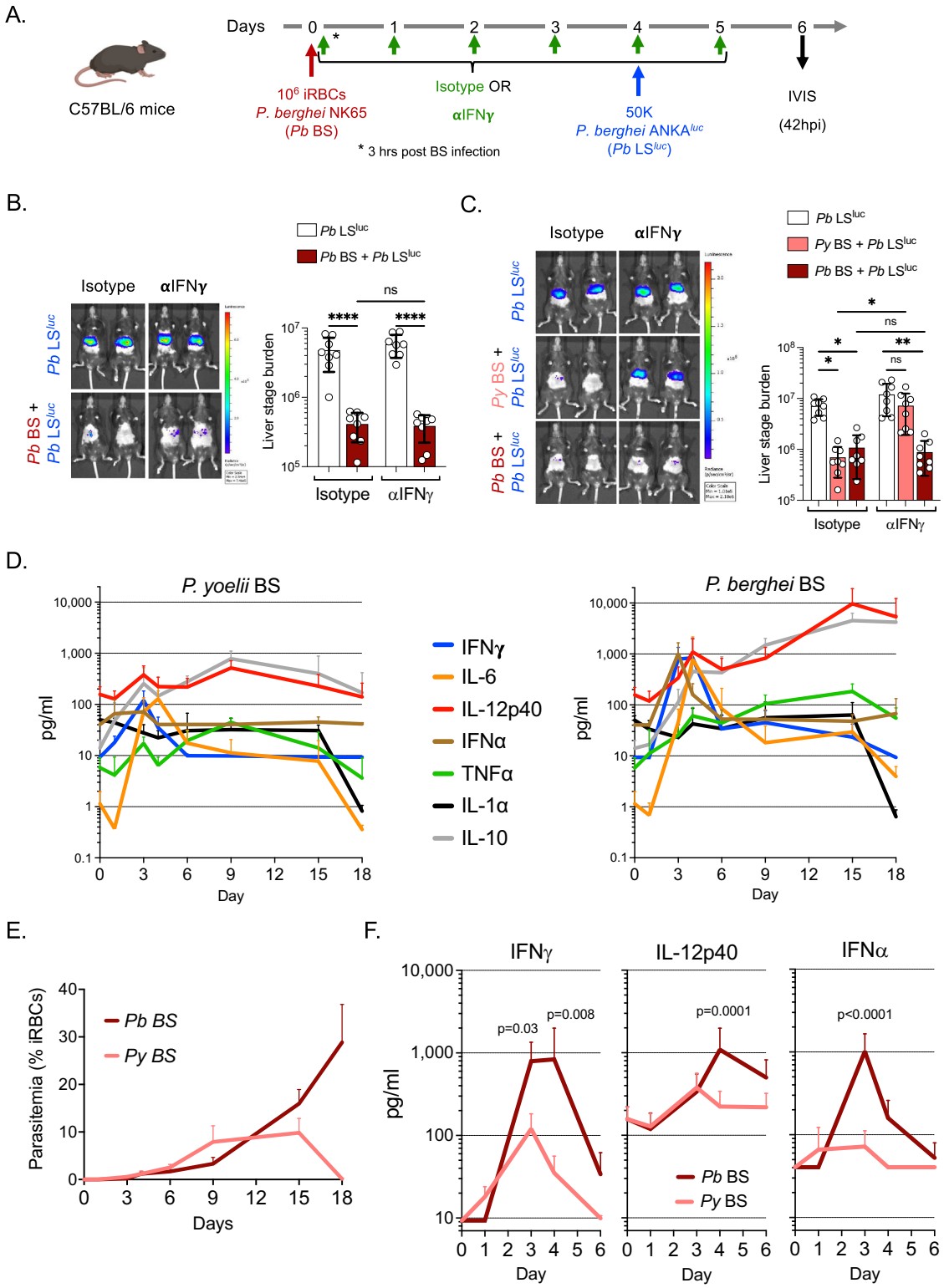

IFNγ, IL-12p40, and IFNα cytokines in *Pb* BS-infected mice compared to *Py* BS-infected mice particularly between days 3–6 post BS infection when the parasitemia between two parasite infections were similar (Fig. 5E, F). Importantly, *Py* BS-infected C57BL/6 mice did not show pronounced induction of IL-12p40 and IFNα. The poor induction of these inflammatory cytokines was also observed in *Py* BS-infected Balb/c mice (Fig. S3). In contrast, we observed 10- to 20-fold increases in expression of IL-12p40 and IFNα in *Pb* BS-infected mice (Fig. 5F). These data show that *Pb* BS infection provokes distinct host responses

when compared to *Py* BS infection and suggests that IL-12p40 and IFNα might contribute to *Pb* BS-mediated suppression of LS development.

We therefore assessed if IL-12p40 or IFNα individually play a role in suppression of LS development during concurrent *Pb* BS infection in C57BL/6 mice. Blocking IFNAR1 signaling (Fig. 6A) or neutralizing IL-12p40 (Fig. 6B) individually in *Pb* BS + *Pb* LS^luc infected mice did not restore LS development. Thus, we next neutralized IL-12p40, IFNγ, and blocked IFNAR1 signaling in dual combinations. Neutralizing IL-12p40 with blockade of IFNAR1 (Fig. 6C) or neutralizing IL-12p40

**Fig. 5 | *P. yoelii* blood stage and *P. berghei* blood stage infections provoke distinct host responses that suppress liver stage development. A** Experimental layout. Control (*Pb* LS^luc) and *Pb* NK65 BS-infected (*Pb* BS + *Pb* LS^luc) C57BL/6 mice were treated with isotype or anti-IFNγ and 4 days later were infected with 50,000 *Pb* ANKA^luc sporozoites. **B** Parasite liver stage burden was measured by IVIS at 42 hpi and represented as total flux (p/s). *n* = 8 mice per group. ****$P < 0.0001$. **C** Control (*Pb* LS^luc) and mice infected with $10^6$ *Py* XNL iRBCs (*Py* BS + *Pb* LS^luc; pink bars) or *Pb* NK65 iRBCs (*Pb* BS + *Pb* LS^luc; dark red bars) were treated with isotype or anti-IFNγ as described above and four days later were infected with 50,000 *Pb* ANKA^luc sporozoites. Parasite liver stage burden was measured by IVIS at 43 hpi. *n* = 8 mice per group. *$P < 0.05$, **$P = 0.003$. **D** C57BL6 mice were infected with $10^6$ *Py* XNL or *Pb* NK65 iRBCs. Blood was collected at different time points during the course of

infection and subjected to comparative cytokine kinetic analysis by LEGENDplex multi-analyte flow assay. The expression of the cytokines was represented as pg/ml. *n* = 6 mice per group. **E** Blood stage parasitemia during the course of *Py* or *Pb* BS infection in C57BL/6 mice. *n* = 6 mice per group. **F** Comparison of the expression of IFNγ, IL-12, or IFNα between *Py* or *Pb* BS infection. *n* = 6 mice per group. **B, C** Two-way ANOVA with Tukey's multiple comparison test for comparing groups with two variables. **F** Two-way ANOVA with Šídák's multiple comparison test for comparing individual time points between the groups. Results are combined and represented as means ± SD from two independent experiments. Source data are provided as a Source data file. The graphical illustration of the mouse in (**A**) was made using BioRender.com.

together with IFNγ (Fig. 6D) did not restore *Pb* LS development during concurrent *Pb* BS infection. Strikingly however, neutralization of IFNγ together with blockade of IFNAR1 substantially restored *Pb* LS development during concurrent *Pb* BS infection (Fig. 6E). These results suggest that *Pb* BS infection suppresses *Pb* LS development via both type I and II interferons. Lastly, we determined that *Pb* BS-induced type I and II interferons also suppresses *Py*GAP LS development in Balb/c mice (Fig. S4).

## Discussion

*Plasmodium* parasites progress through their life cycle by first infecting the mammalian liver, where the parasites initiate and complete LS development to form the first generation of merozoites, which initiate BS infection. Since this life cycle progression is unidirectional, crosstalk between BS and LS via host responses appears to be of little consequence. However, during natural parasite infection in high transmission malaria-endemic regions, where individuals are potentially exposed to hundreds of infected mosquito bites either seasonally or throughout the year[29], the likelihood of harboring LS and BS infection simultaneously is substantial. In fact, many individuals in endemic regions already carry BS parasite infection, while getting exposed to subsequent new infections[30,31]. Thus, it is important to understand how the interplay between an existing BS infection and a subsequent LS infection might impact the courses of natural infections and the likelihood of super-infection with different parasite strains. This is also relevant in the context of immunization with live-attenuated replication-competent parasite vaccines, as it could reduce vaccine efficacy in BS-infected vaccinees.

Our data show that *Plasmodium* BS infection suppresses concurrent LS infection in different mouse strains infected with different parasite species. We also demonstrate that the impact of BS infection-induced host responses manifests during LS development but not during the early events of hepatocyte infection such as invasion by sporozoite stages and intracellular transformation of sporozoites into LS trophozoites. Interestingly, it was previously shown that hepatocytes harvested from BS-infected mice failed to support normal LS infection, suggesting that the BS-induced host responses condition hepatocytes to become refractory to LS development[32]. Importantly, the data presented herein refute the previously claimed role for the iron-regulatory hormone hepcidin in BS-mediated suppression of LS development and by extension, a role for hepcidin in shaping clinical malaria epidemiology in endemic regions[5,7]. We show that hepcidin has no role in BS-mediated suppression of LS development using an attenuated *Py* GAP[11] and wild-type parasites, including the *Pb* infection model used in the previous study[5,6]. Hepcidin is a master regulator of systemic iron homeostasis that prevents the iron overload-mediated toxicity in organs by maintaining iron equilibrium[33]. During infection, inflammation-driven hepcidin restricts iron availability to inhibit pathogen growth[34]. Yet, we show that maintaining hepcidin levels at baseline during BS infection did not prevent BS-mediated suppression of LS development. Our data thus strongly suggest that high hepcidin levels during ongoing blood stage infection might not protect humans from subsequent *Plasmodium* infection as previously claimed[5–7]. The

role of hepcidin in this context has also been brought into question by a longitudinal clinical study of children living in a malaria endemic area, which found no association between high hepcidin levels and a reduced risk of subsequent *P. falciparum* malaria[35].

Our data does reveal that BS infection-induced interferons, namely IFNγ and type-I IFN signaling, mediate suppression of wild-type parasite as well as attenuated parasite LS development[11]. This again stands in contrast to previous findings[5,6]. We show that the expression of these cytokines varies during BS infections with different parasite species. While we found that IL-6 and IFNγ were highly elevated from their baseline levels during both *Py* and *Pb* BS infection, IL-12p40 and IFNα were elevated only in response to *Pb* BS infection. Interleukin-6 has previously been shown to strongly suppress LS infection[22,23]. Yet, in our study, neutralizing IL-6, which is also responsible for the induction of hepcidin, could not prevent the BS-mediated suppression of LS development. In contrast, neutralizing IFNγ largely restored normal LS development of both *Py* and *Pb* parasites during ongoing *Py* BS infection. Our data show that IFNγ does not affect the process of sporozoite invasion of hepatocytes or the establishment of LS trophozoites but subsequently suppresses LS development. Importantly, IFNγ pre-treated hepatocytes were unable to support normal LS development, which suggests that IFNγ-signaling renders hepatocytes refractory to support LS development, without continuous exposure. This finding re-ascertains previously made observations[32]. IFNγ activates JAK/STAT1 (Janus-activated kinases/signal transducer and activator of transcription 1) signaling pathways by phosphorylating cytosolic STAT1[36]. This signaling is essential for the induction of highly reactive nitric oxide (NO) and for triggering autophagy[37,38], both mechanisms reported for IFNγ-mediated suppression of LS infection within hepatocytes[32,39–42].

We further found that while neutralizing IFNγ alone did not rescue LS development during ongoing *Pb* BS infection, concomitant blocking of type-I IFN signaling together with neutralizing IFNγ rescued LS development. This indicates that signaling pathways within hepatocytes for both interferons are triggered by *Pb* BS infection and simultaneously suppress LS development. The type-I (IFNα and IFNβ) and type-II (IFNγ) interferons are known to activate common as well as distinct STAT pathways which regulate hundreds of genes downstream that mediate various cellular responses[36,43]. Importantly, both IFN types commonly facilitate the phosphorylation and dimerization of STAT1[44,45]. This could be the reason why the individual blockade of either of these interferons alone during *Pb* BS infection did not reverse the suppressive effect on LS development.

We have previously shown that immunizations with replication-competent whole parasite vaccines engender unprecedented stage- and species-transcending protection against infection in rodent malaria models[9,46]. This superior protection can be attributed to the diversity of antigen expression and increase in cell mass during intra-hepatocytic LS immunogen developement[9,46]. Reducing the replication potential of such vaccines is thus likely to affect their efficacy. Indeed, clinical trials conducted with *P. falciparum* replication competent wild-type parasite vaccination exhibited different efficacy,

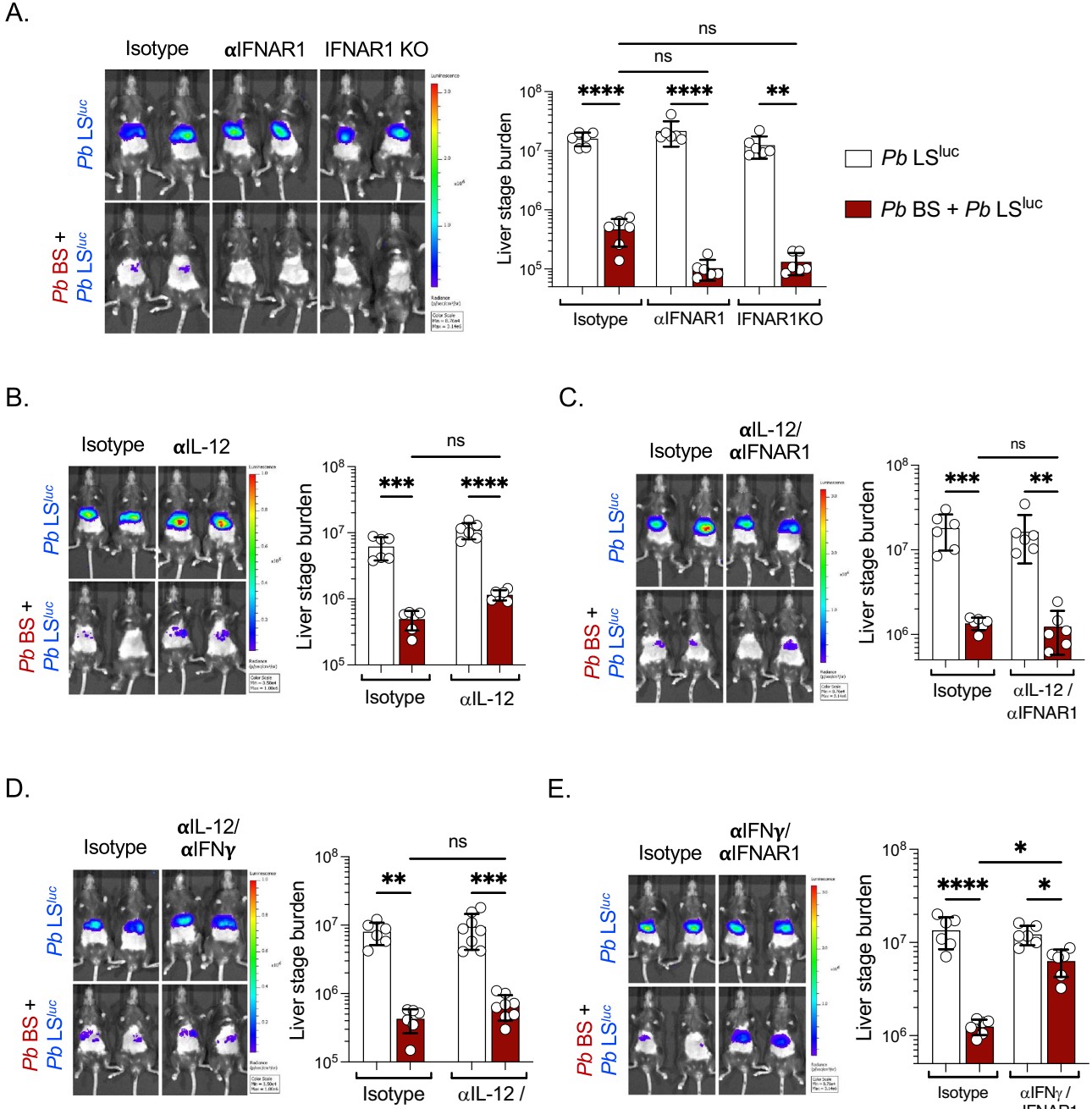

**Fig. 6 | *P. berghei* blood stage infection suppresses liver stage development via both type I and II interferons.** Wild type or IFNAR1$^{-/-}$ C57BL/6 mice were infected with 10$^6$ *Pb* NK65 iRBCs (*Pb* BS + *Pb* LS$^{luc}$). *Pb* BS-infected and Control (*Pb* LS$^{luc}$) mice were treated with isotype controls or **A** anti-IFNAR1 (*n* = 6 mice per group; **P* = 0.001, *****P* < 0.0001), **B** anti-IL-12 (*n* = 6 mice per group; ****P* = 0.0003, *****P* < 0.0001), **C** anti-IL-12/anti-IFNAR1 (*n* = 6 mice per group; ***P* = 0.002, ****P* = 0.0009), **D** anti-IL-12/anti-IFNγ (*n* = 6–8 mice per group; ***P* = 0.001,

****P* = 0.0003), or **E** anti-IFNγ/anti-IFNAR1 (*n* = 6 mice per group; **P* < 0.05, *****P* < 0.0001) monoclonal antibodies. Four days later, the mice were infected with 50,000 *Pb* ANKA$^{luc}$ sporozoites. Parasite liver stage burden was measured by IVIS at 42–43 hpi and represented as total flux (p/s). Two-way ANOVA with Tukey's multiple comparison test for comparing groups with two variables. Results are combined and represented as means ± SD from two independent experiments. Source data are provided as a Source data file.

depending on dosing schedules. Presumably, this was due to the presence or absence of BS parasites during vaccine administration[10,47]. Specifically, subjects were administered with three doses of 5.12 × 10$^4$ infectious sporozoites under chloroquine drug cover (PfSPZ-CVac) with either five[47] or seven[10] day intervals between each dose. Strikingly, while vaccination using five-day intervals induced 63% sterile protection against challenge[47], vaccination using seven-day intervals showed no protection[10]. Chloroquine treatment allows completion of the

6.5–7-day-long *P. falciparum* LS development[48], and also does not affect red blood cell invasion by exo-erythrocytic merozoites and early intra-erythrocytic infection but kills replicating BS parasites[49]. Therefore, administration of the 2nd and 3rd PfSPZ-CVac doses in the 7-day interval regimen apparently coincided with the transient BS infection caused by the prior PfSPZ-CVac administration[10]. Since, there was no difference in the vaccine dose or number of immunizations[10,47], the presence of BS infection during vaccination appears to be the only

factor that was associated with the loss of vaccine-mediated protection[10]. We have previously shown that type-I IFN signaling during replication-competent whole sporozoites vaccination impairs hepatic CD8+ T cells, which are critical for conferring protection against LS infection[50]. Additionally, IFNγ responses were shown to promote the contraction of antigen-specific CD8+ T cell responses and limit the formation of memory cell populations in other infection models[51,52]. These studies collectively indicate that the BS-induced interferons might not only mediate suppression of LS development but might also interfere with the process of inducing vaccine-mediated adaptive immunity. Based on these findings and data presented herein, clearing BS parasitemia and more importantly normalizing the inflammation prior to vaccination in malaria endemic regions might significantly enhance immunogenicity and efficacy of replication-competent whole parasite vaccines. Our findings warrant future exploration into how concurrent BS infection-induced inflammatory responses in humans might affect vaccine-mediated protective adaptive immune responses.

Finally, in malaria endemic regions, young children were shown to have high IFNγ responses to *P. falciparum* BS infection[53]. Notably, these IFNγ responses were associated with resistance to re-infections[54–56]. Thus, it is tempting to hypothesize that the IFNγ-mediated suppression of LS development might lower subsequent *Plasmodium* infection rates, thereby reducing infections with different parasite strains in the same individual. This might lower the incidence of severe clinical malaria[57], and to some extent might explain malaria epidemiological patterns as postulated in previous studies[5,7].

## Limitation of the study

Our study reveals how BS-induced innate inflammation during ongoing infection severely impacts LS development. However, we did not study whether after cessation of BS infection, these IFN responses persist and if their impact on hepatocytes could still suppress LS development. Our data show that the BS-induced IFNs remain high over their baseline levels between days 3–8 during the course of infection. The decline of IFN responses after day 8 might reduce the suppression of LS development. However, adaptive immune responses that are engendered by BS infection might then suppress LS infection after innate responses have subsided. In fact, it was shown that mice infected with non-lethal rodent malaria BS parasites generate adaptive immune responses to pre-erythrocytic stages[58]. Therefore, it would be difficult to distinguish between the role of BS-induced innate vs adaptive immune responses in suppressing LS infection after cessation of BS infection. In general, our work was conducted with rodent malaria infection models. While our findings can frame future hypothesis-driven research questions for human malaria infection, any direct interpretation of phenomena observed during human malaria infection must be made with a full acknowledgment that rodent malaria models have limitations, and no interpretation should be considered definitive.

## Methods

### Mice

Six- to eight-week-old female Balb/c, wild type, IFNγ-/- and IFNAR1-/- C57BL/6 mice were purchased from The Jackson Laboratory and female Swiss Webster (SW) mice for sporozoite production were purchased from Envigo laboratories. The mice were maintained under pathogen-free conditions with 12 h light/12 h dark cycle, 72°F temperature and 45% humidity at the Center for Global Infectious Disease Research, Seattle Children's Research Institute (SCRI). Food and water were provided ad libitum to the animals. Animal sex was not considered in study design. All animal procedures were performed as per the regulations of the SCRI's Institutional Animal Care and Use Committee (IACUC). The animal procedures were approved by IACUC under protocols 00507 and 00666.

### Sporozoite isolation

Eight- to ten-week-old SW mice were intraperitoneally (i.p.) injected with luciferase-expressing *P. yoelii* LS replication-competent genetically attenuated parasite (*Py* GAP*luc*), *Py*GFP-luc, *Py* XNL, *Py*UIS4+GFP or *P. berghei* ANKA (*Pb* ANKA*luc*) infected blood. Gametocyte exflagellation was confirmed 3 to 4 days later and infected mice were then used for feeding female *Anopheles stephensi* mosquitoes. Sporozoites were isolated from mosquito salivary glands on day 15 (for *Py* parasites) or 25 (for *Pb* parasites) after the infected blood feed and used for the LS infections[59–62].

### *Plasmodium* blood infection

Primary blood-stage (BS) infection in mice was achieved by intravenous inoculation of $10^5$ or $10^6$ *P. yoelii* XNL or *P. berghei* NK65 (non-lethal strains) infected RBCs (iRBCs), respectively. The parasitemia in the peripheral blood was determined by microscopic counting of iRBCs in Giemsa-stained thin blood smears and represented as % iRBCs.

### *Plasmodium* liver infection

Mice were infected with 50,000 sporozoites by intravenous route of injection. Parasite liver load was quantified at 41–43 h post-infection by measuring bioluminescence through real-time in vivo imaging with IVIS Lumina Imaging System (Caliper Life Sciences, USA)[63]. Briefly, mice were injected i.p. with 150 µl of 15 mg/ml D-luciferin (Gold Biotechnology) and were kept under anesthesia for 5 min using isoflurane-anesthesia system (XGI-8, Caliper Life Sciences, USA). Bioluminescence imaging was acquired while the mice are still under anesthesia by keeping instrument settings to medium binning factor, 10 cm FOV and the exposure time to 3 min. The analysis was done using Living Image® 4.0 software (v4.3.1.0.15766) by measuring the luminescence signal intensity as total flux (p/s) while selecting the region of interest (ROI) around the abdominal area at the location of the liver.

### Quantification of Hepcidin

*By ELISA*. The blood was collected retro-orbitally in heparin tubes from naïve or BS parasite-infected mice on day 4 or 5 post infection. The plasma was harvested and circulating hepcidin was measured by competitive binding ELISA according to the manufacturer's protocol (Intrinsic Life Science, USA)[64]. The standard curve was prepared by 3-fold dilutions (8-points) of synthetic hepcidin (provided in the kit) starting at 1000 ng/ml which was then used to interpolate the test samples' absorbance readings that were collected on BioTek microplate reader (ELx800) using Gen5 software (v2.06.10). The obtained values were multiplied with the dilution factor to get the final concentration of hepcidin (ng/ml) in the blood. *By qPCR*. On day 6 or 7 post BS stage infection, ~50 mg of liver tissue was collected and homogenized into Qiazol (Qiagen) solution. Total RNA was extracted using miRNeasy kit (Qiagen #217004). The complementary DNA (cDNA) was synthesized from 1 µg of total RNA using QuantiTect Reverse Transcription Kit (Qiagen #205311) and quantitative PCR (qPCR) was performed using SYBR Green Master Mix (Bimake #B21202) on QuantStudio 5 Real-Time PCR system[65]. Briefly, 0.5 µL of diluted cDNA (1:50) was used in total 10 µL reaction volume to amplify liver *Hamp* gene using forward primer 5′-AAGCAGGGCAGACATTGCGAT-3′ and reverse primer 5′-CAGGATGTGGCTCTAGGCTAT-3′, and the mouse *Gapdh* gene as an internal control, using forward primer 5′-CCTCAACTACATGGTCTACAT-3′ and reverse primer 5′-GCTCCTGGAAGATGGTGATG-3′. The expression of *Hamp* mRNA was calculated by comparative CT analysis method $(2^{(-\Delta\Delta CT)})$ and represented after normalizing it to the control (uninfected) group[66].

### Antibodies and LDN193189

Mice infected with BS parasites and control mice were injected i.p. with 0.4 mg of anti-IL-6 (clone MP5-20F3; BioXcell# BE0046) alone, LDN193189 (3 mg/kg; Sigma-Aldrich #SML0559) alone, anti-IL-6/LDN193189 combined, 0.4 mg of anti-IFNAR1 (clone MAR1-5A3; Leinco

Technologies #I-401), 0.4 mg of anti-IFNγ (clone XMG1.2; Leinco Technologies #I-1119), 0.4 mg of anti-IL-12 (clone C17.8; Leinco Technologies #I-1175) alone or in combinations or Isotype control monoclonal antibodies every day starting from day 0 (3 h after inoculation with iRBCs) to 5 or 6 days after BS infection.

## Hepatoma cells maintenance, treatment, and infection

HepG2-CD81 hepatoma cells (originally obtained from Olivier Silvie Laboratory, Cimi, Paris, France) were cultured and maintained in DMEM medium supplemented with FBS, GlutaMAX and pen/strep antibiotics. For live imaging, 50,000 cells per well were seeded in 96-well plate (Greiner Bio-one #655077) 30 h before sporozoite infection. For immune-fluorescent assay (IFA), 35,000 cells per well were seeded in 18-well microchamber slides (Ibidi #81816). HepG2-CD81 cells were treated with 10 U/ml of human recombinant IFNγ at 16 h, or 3 h pre-infection or during the sporozoite addition to HepG2-CD81 cells for up to 3 hpi. For sporozoite invasion assay, the HepG2-CD81 cells were infected with *Py* XNL parasite strain. For the analysis of developed LS at 48 hpi, cells were infected with *Py*^UIS4-GFP transgenic line. Infection was conducted with 1/5 MOI.

## Immunofluorescence assay (IFA) and microscopy

*Invasion assay.* Cells infected with *Py* XNL sporozoites were fixed and treated for IFA at 6 hpi as described previously[67]. Briefly, cells were fixed with 4% v/v paraformaldehyde (PFA) and permeabilized with 0.5% v/v Triton X-100 solution in 1x PBS for 10 min. Cells were then incubated with anti-*P. yoelii* UIS4 antibody (1:200; gifted by Photini Sinnis laboratory, JHU, USA) followed by staining with a secondary anti-rabbit fluorescent antibody conjugated with Alexa-fluor 594 (1:1000; Fisher scientific #A11012). Productive invasion was scored on Stellaris-8 confocal microscope (Leica Microsystem, Deerfield, IL, USA) by counting UIS4 positive sporozoites and micrographs were taken using a HC PL APO 63x lens. Three independent experiments were conducted. *Liver stage development* (in vitro). Cells infected with *Py*^UIS4-GFP sporozoites were fixed and permeabilized as described above at 48 hpi. Cells were then incubated for 1 hour with phalloidin rhodamine (1:200; Fisher Scientific #R415) to label f-actin and Hoechst 33342 (1:100; Fisher Scientific #H3570) to visualize host and parasite DNA. The infectivity was determined by counting the number of liver stages after imaging on Stellaris-8 confocal microscope using a HC PL APO 20x lens. Image analysis was further done to quantify the size of 100 liver stages per condition using LAS-X (Leica Microsystem) software. Three independent experiments were conducted. *Liver stage development* (in vivo). Livers were perfused with 1×PBS, fixed in 4% PFA in 1×PBS and lobes were cut into 50 μm sections using a Vibratome apparatus (Ted Pella, Redding, CA). The liver sections were permeabilized in 1×TBS containing 3% v/v H$_2$O$_2$ and 0.25% v/v Triton X-100 for 30 min at room temperature. Sections were then blocked in 1x TBS containing 5% v/v dried milk (TBS-M) for at least 1 h and incubated with anti-*P. yoelii* UIS4 antibody (1:500; gifted by Photini Sinnis laboratory, JHU) primary antibody in TBS-M at 4 °C overnight. After washing with 1x TBS, the sections were incubated with fluorescent secondary antibody, donkey anti-rabbit Alexa Fluor 488 (1:1000; Thermofisher Scientific #A-21206) and 1 μg/ml 4′,6-diamidino-2-phenylindole (DAPI) in TBS-M for 2 h at room temperature. The sections were washed and incubated in 0.06% w/v KMnO$_4$ for 30 s to quench background fluorescence. Sections were then again washed and mounted on the glass slide with FluoroGuard anti-fade reagent (Bio-Rad, Hercules, CA). The images were acquired on Stellaris-8 confocal microscope using a HC PL APO 63x lens. The images were processed using LIGHTNING mode on the LAS-X (Leica Microsystem) software. For quantification of the size, LS parasites were assumed to be elliptical in shape and area was calculated from its longest (a) and shortest (b) circumferential diameter (πab). At least 15 parasites were counted for each mouse from each group. Two independent experiments were conducted.

## Multiplex cytokine assay

Mice were infected with BS parasites by injecting $10^6$ iRBCs of either *Py* or *Pb* parasite. Blood was collected from heparin tubes at different time points during the course of infection and plasma was harvested by spinning tubes at 10,000 rpm at 4 °C for 10 min. The samples were immediately stored at −80 °C until further use. The expression of cytokines (IFNγ, IL-6, IL-12p40, IFNα, TNFα, IL-1α, IL-10, IFNβ, IL-1β, IL-12p70, and TGFβ) was determined in bead-based multiplex immunoassay using a customized LEGENDPlex mouse cytokine panel (Biolegend, San Diego, USA). The assay was performed as per manufacturer's instructions and data was acquired on BD LSR II flow cytometer using BD FACSDiva software (v8.0.1). The data was analyzed using LEGENDplex™ Data Analysis Software, (v2023-02-12, 58444) (BioLegend, USA) as shown in Fig. S3A[68]. The standard curve was created for each analyte ($R^2 > 0.99$) by performing 1:4 dilution (8-points) of mouse custom standard panel (provided in the kit) with starting top concentration 10,000 pg/ml or 50,000 pg/ml (IFNβ). The assay was performed by simultaneously using the samples from two to three independent experiments.

## Statistical analysis

The experiments were performed using the number of mice between 3–12 per group. The presented data are either from one representative experiment of three independent experiments with similar results obtained or combined from two to three independent experiments. The statistical analyses were performed in GraphPad Prism v10. The data were analyzed using two-sided non-parametric Mann–Whitney U-test for two groups data comparison, Kruskal–Wallis test followed by Dunn's multiple comparison test for the data involving three or more groups, one-way or two-way ANOVA with Tukey's multiple comparison test for comparing groups, or two-way ANOVA with Šídák's multiple comparison test for comparing individual time points between the groups. A *p* value of <0.05 was considered as statistically significant.

## Reporting summary

Further information on research design is available in the Nature Portfolio Reporting Summary linked to this article.

## Data availability

All pertinent data generated or analyzed during this study are included in this published article and its accompanying supplementary information files. The source data underlaying each figure are provided as a source data file. Source data are provided with this paper.

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

## Acknowledgements
We thank the staff of the vivarium at Seattle Children's Research Institute for their constant support of the animal studies presented here. In addition, we thank the insectary staff for their diligent work in rearing the mosquitoes for these studies. This work was funded by Seattle Children's internal seed funds to S.H.I.K.

## Author contributions
H.P. and S.H.I.K. conceived the study. H.P., N.K.M., S.K., G.Z., A.S., D.G., J.A., B.A.A., W.B., L.R., N.C., and A.A.S. planned and performed the experiments. H.P., A.S., and D.G. performed data analysis. H.P. and S.H.I.K. performed data visualization. S.H.I.K. supervised and acquired the funding. H.P. and S.H.I.K. wrote the original manuscript draft. All authors were involved in reviewing and editing the final manuscript.

## Competing interests
The authors declare no competing interests.
