## [Peer Review File · Nature Communications]

Malaria blood stage infection suppresses liver stage infection via host-induced Interferons but not hepcidinReviewers' Comments:

Reviewer #1:

Remarks to the Author:

This manuscript from Patel et al attempts to show how blood-stage malaria would negatively impact concurrent liver-stage infection. Understanding the mechanism of this process would allow us to inhibit or reverse this process in the context of live-attenuated sporozoite-based vaccination against malaria. It is important that live-attenuated sporozoites survive in the liver to the fullest extent permitted by the genetics of the vaccine strain to induce strong immune responses and possibly protection. The most important finding of this manuscript may be that IFN γ may have a role in limiting LS malaria and that it may be driven by BS malaria. However, there is existing literature indicating that BS malaria can alter immune responses and impact and control of a variety of other infections, including viral infections. Therefore, BS malaria limiting a new infection, albeit it being malaria itself, is not surprising.

While the intent of this manuscript is commendable, the overall findings are somewhat preliminary, the scope is limited, and some of the key claims are not supported experimentally. There appear to be technical and experimental caveats in some of the key studies. Please see below:

1. The scope of this manuscript is narrow, partly owing to the examination of the short time span after the inception of blood-stage malaria. The authors would need to determine the kinetics of IFN responses generated in BS malaria and assess for how long after the initiation of BS malaria would it impact the control of LS malaria. The prediction that the epidemiological patterns observed in malaria-endemic regions may depend on the proposed phenomenon is unsupported in the current scope of the experiments and somewhat premature without understanding the timeline of how BS malaria can impact LS malaria.
2. In lines of the above point, the manuscript generally falls short in providing definitive evidence for its claims. For example, in lines 98-99, the authors claim that IFN γ mediates a developmental suppression of wild-type LS infection. There is no data indicating that IFN γ suppressed the development of the parasite in the liver (as opposed to the invasion, immune-mediated elimination etc.). The authors will need to determine if IFN γ suppresses the development of Plasmodium in the hepatocytes.
3. The authors use IL-6 blockade and LDN treatment to inhibit hepcidin, to conclude that the latter does not impact the development of Plasmodium in the liver. They have not determined the possibility of IL-6 blockade directly impacting liver-stage malaria, nor the potential time scale of such an impact. No analysis is provided to determine this in Figure 1E, albeit it being just at a single time point.
4. Antibody-mediated blockade of IFNAR is dose-dependent, as is the case with IFN γ neutralization. With increasing levels of IFN γ or IFNAR-expressing cells, the dose of blocking antibodies required would increase- temporally, further complicated by its local bioavailability. More conclusive and reliable data would be generated using receptor deletion approaches. IFN γ KO, IFNARKO, IFNAR1fl/fl mice, etc., are readily available and some of these strains have been used by the authors in the past. The authors should resort to such precise approaches, especially considering their findings are not in agreement with other previously published data.
5. The key statistics in Fig S1F comparing the 3rd and 4th group are missing. Control wild-type mice for Fig 2D are missing.
6. Considering the timeline of the experiments, the IFN γ produced by innate immune responses to BS malaria would be expected to impede LS malaria. Nevertheless, this is not a certainty and the extent to which innate and adaptive responses elicited by BS malaria would drive the phenotype observed would expand the scope of the manuscript.

Reviewer #2:

Remarks to the Author:

This study shows that 1, Concurrent blood stage (BS) infection inhibits liver stage development; 2, IFN-g is critical in controlling liver stage (LS) development in two rodent malaria parasite species; 3, Hepcidin has no role in the inhibition of LS development. The observations are interesting and may provide important information for vaccine development. It has been reported that blood-stage infections can hinder the development of protective responses to malaria in a PfSPZ-CVac efficacy study. There are also plenty of reports showing the protective roles of IFN-g in malaria. This study provides data suggesting that the lack of protection in the PfSPZ-CVac efficacy study could be due to increased IFN-g levels induced by BS.

The experiments clearly show that IFN-g is critical in controlling LS development. However, it is possible the IFN-g needs to function with other cytokines/chemokines and immune cells to achieve the inhibition of LS. Injection of IFN-g before LS infection may provide some answers to this question.

High levels of IFN-g can be induced by infections with other pathogens such as viruses and bacteria. The authors may want to infect mice with a virus or another parasite to induce IFN-g before LS infection to test this possibility.

Another issue is strain-specific immunity that has been demonstrated in malaria parasite infections. If it is only the IFN-g level that affects SPZ vaccine efficacy, strain-specific immunity should not be an issue. It would be nice to infect the host with one strain of *P. yoelii* BS and then infect with LS of another *P. yoelii* strain or *P. berghei*.

Line 103-107: If the high level of IFN-g can suppress the growth of vaccine sporozoites, then it can also protect the child from natural infection. In this case, why do we need to vaccinate people with BS infection?

Line 108-111. It would be nice to include experiments after curing active BS infection before liver stage infection.

As a paper for nature Communications usually requires elucidation of the mechanism, it would be nice to present data showing how IFN-g inhibits LS growth.

Malaria parasite infection also induces immune inhibition including inhibition of many T and B cell pathways. The suppression of the immune system by prior infection may also affect antibody responses even sporozoites can develop normally. The lack of antibody response after vaccination may be due to a general mechanism of immune inhibition. The inhibition can affect vaccine efficacy for all types of vaccines including molecular vaccines without the need for LS development.

Reviewer #3:

Remarks to the Author:

The manuscript by Patel et al describes that infection initiated by Py and Pb blood stages inhibits subsequent liver infection by sporozoites (~100-fold decrease in parasite load by bioluminescence in Py and ~10-fold decrease in Pb).

This inhibitory phenotype was first characterized by Portugal et al 2011 and the level of inhibition correlated with the increase in hepcidin levels in PbNK65 infected mice. Over-expression of hepcidin in naïve mice decreased liver infection by ~50% (~2-fold). Based on these results Portugal et al suggested that up-regulation of hepcidin during blood infection inhibited liver infection by sporozoites. Notably lack of IFN-g in knockout mice could not revert this inhibitory phenotype.

On the contrary, in the submitted manuscript, Patel et al show that anti-IFN γ antibodies and lack of IFN γ in knockout mice can revert the liver infection inhibition phenotype caused by Py XNL blood infection. Additionally, they could not revert liver infection inhibition in blood infected mice by blocking the up regulation of hepcidin using anti-IL6 antibodies + LDN inhibitor. Finally, the authors conclude that "malaria blood stage infection suppresses liver stage infection via IFN γ but not hepcidin". This is an important article that challenges the actual model of liver infection inhibition by hepcidin in blood infected mice with potential consequences for live attenuated sporozoite vaccination as well as for better understanding the dynamics of multiple re-infections in the field.

However before recommending this article for publication, I would like to clarify some points.

- 1- In this manuscript, the authors use a mutant parasite PyGAPLuc displaying a late growth impairment to measure "liver infection". They generalize the conclusion of their findings regarding the effect of IFN γ using this mutant - as can be seen in the title of the manuscript. An experiment using PyXNLluc sporozoite infection should be performed to confirm that the role of IFN γ is not associated with the PyGAPLuc phenotype.
- 2- Given the fact that Ifng KO mice did not revert the phenotype of liver inhibition in Portugal et al 2011, it would be judicious to test anti-IFN γ antibodies and Ifng KO mice using PbNK65 blood infection and PbANKAluc sporozoite challenge as performed in the supplementary figure 1 for anti-IL6/LND inhibitor treatment. This experiment will also provide evidence that the IFN γ role is parasite species transcendent, and it is not only valid for Py.
- 3- It is important to measure the serum concentration of IFN γ in the Py and Pb experiments like the group did in Miller et al, 2014, to verify if they correspond to the concentration required to block liver infection, as well as to confirm the IFN γ -/- phenotype of KO mouse. Equally, parasitemia should be controlled such as in the experiments using anti-IL6/LND inhibitor.
- 4- In figure 2D, there is almost a 5-fold difference between control and infected IFN γ KO mice, this is almost the difference found in Pb (SFig. 1E). Nonetheless, the figure shows a non-significant difference. Could you please check the statistics? The mean and SD suggest that these values are statistically different.
- 5- To investigate the effect of hepcidin on Py sporozoite liver infection, the authors sought to inhibit the signaling pathways responsible for hepcidin induction using anti IL-6 blocking antibodies and LDN193189. They report the effect of the treatment on serum hepcidin level and parasitemia at day 6 post-infection and treatment while Py sporozoite challenge is performed at day 4 post-infection. The authors should show the level of hepcidin at the day of sporozoite challenge as presented for the experiment using Pb.

Minor comments:

- 6- The authors convincingly demonstrate that sporozoite infection in mice exhibiting concurrent blood stage infection results in reduced hepatic parasite burden at 40 h post-infection. Analysis at earlier time points is not presented here and an impact of blood stage infection on sporozoite homing to the liver, crossing into the hepatic parenchyma or phagocytosis in the bloodstream therefore cannot be ruled out. Sentences stating that blood stage infection suppresses parasite development (line 54 and line 56) should therefore be reformulated to take into consideration this possibly broader impact on liver stage infection (e.g. by substituting "LS development" by "liver infection", as in the title of the manuscript).
- 7- Could the authors elaborate on why different inocula of Py and Pb iRBCs were used and why the timings of sporozoite infection were different?
- 8- In the material and methods section, it's unclear which doses of anti-IFN gamma and anti-IFNAR antibodies were administered to the mice to abrogate signaling through these pathways
- 9- In Supplementary Fig 1, statistical significance is not indicated on all figures, symbols are missing above some of the bars.
- 10- In Supplementary Fig 2B, the mention of which of the two quantifications refers to the group treated with isotype or anti-IFNAR antibodies is missing on the graph.

Response to Reviewers

We like to thank reviewers for their positive comments and constructive critiques, which have helped us to strengthen the manuscript. As we agreed with the editor, we are now submitting the revised manuscript as a full length research article due addition of a substantial new data to address the concerns raised by the reviewers. The manuscript has been restructured to the research article format. Below, please find the point-by-point responses to each reviewers' critique.

Reviewer #1:

This manuscript from Patel et al attempts to show how blood-stage malaria would negatively impact concurrent liver-stage infection. Understanding the mechanism of this process would allow us to inhibit or reverse this process in the context of live-attenuated sporozoite-based vaccination against malaria. It is important that live-attenuated sporozoites survive in the liver to the fullest extent permitted by the genetics of the vaccine strain to induce strong immune responses and possibly protection. The most important finding of this manuscript may be that IFN γ may have a role in limiting LS malaria and that it may be driven by BS malaria. However, there is existing literature indicating that BS malaria can alter immune responses and impact and control of a variety of other infections, including viral infections. Therefore, BS malaria limiting a new infection, albeit it being malaria itself, is not surprising.

While the intent of this manuscript is commendable, the overall findings are somewhat preliminary, the scope is limited, and some of the key claims are not supported experimentally. There appear to be technical and experimental caveats in some of the key studies. Please see below:

1. The scope of this manuscript is narrow, partly owing to the examination of the short timespan after the inception of blood-stage malaria. The authors would need to determine the kinetics of IFN responses generated in BS malaria and assess for how long after the initiation of BS malaria would it impact the control of LS malaria. The prediction that the epidemiological patterns observed in malaria-endemic regions may depend on the proposed phenomenon is unsupported in the current scope of the experiments and somewhat premature without understanding the timeline of how BS malaria can impact LS malaria.

→ We agree that determining the kinetics of IFN responses generated during BS malaria is important. Therefore, we have now conducted a comparative multi-cytokine kinetic analysis of eleven different analytes including IFN γ during the course of *P. yoelii* (Py) or *P. berghei* (Pb) blood stage (BS) infection. This established a time course and helped to understand how well conserved the BS-induced cytokine responses are that might mediate suppression of LS infection (Fig.5D-F and Fig.S3). Surprisingly, we find differences in responses to BS infection of these parasite species, which we also show, lead to differences in mechanism of suppression.

We would also like to clarify two other aspects raised by the reviewer:

- (i) We are not claiming that epidemiological patterns observed in malaria-endemic regions are solely dependent on the observed impact of BS-induced IFN responses on LS development. We just meant to indicate that the epidemiological impact of hepcidin elevation claimed by the paper of Portugal et al. might rather be due to IFN responses. In addition, evidence from studies done in malaria endemic areas, where children with high IFN γ responses showed more resistance to *P. falciparum* re-infections (Deloron, Chougnet et al. 1991, Luty, Lell et al. 1999, D'Ombrain, Robinson et al. 2008, Robinson, D'Ombrain et al. 2009), support some role of IFN responses in suppression of re-infection. We have now also mentioned the limitations of mouse models in explaining human malaria phenomena in the "limitation of the study" section, as part of the discussion in the revised manuscript.
- (ii) Our analysis show that the BS-induced IFNs remain high over their baseline level between day 3 to 8 during the course of infection. One would assume that a reduction in the IFN responses at later timepoints may lessen the impact of BS infection on LS infection. However, this is difficult to ascertain *in vivo*, since adaptive immune responses that are primed in response to the BS infection will start to play a major role in suppressing the LS infection at later timepoints (Belnoue, Voza et al. 2008). Therefore, interpretation of data obtained from a longer term experiment would be difficult and would not distinguish between the role of BS-induced innate vs adaptive immune responses in suppressing LS infection. We have now mentioned this aspect in the "limitation of the study" section as part of the discussion in the revised manuscript.

2. In lines of the above point, the manuscript generally falls short in providing definitive evidence for its claims. For example, in lines 98-99, the authors claim that IFN γ mediates a developmental suppression of wild-type LS infection. There is no data indicating that IFN γ suppressed the development of the parasite in the liver (as opposed to the invasion, immune-mediated elimination etc.). The authors will need to determine if IFN γ suppresses the development of Plasmodium in the hepatocytes.

→ We agree that this is an important question. We have now performed an *in vitro* infection study and determined that IFN γ does not affect the early phases of hepatocyte infection such as invasion but instead, suppresses LS development (Fig.4). We further show through immunofluorescence assays (IFA) with infected mouse liver sections that neutralizing IFN γ during Py BS infection restores normal development of LS (Fig.S2). Furthermore, we show that neutralizing IFN γ reverses the suppressive effect that BS infection has on LS infection, and we also show that this reversal is observed in IFN γ knockout mice (Fig.3D).

3. The authors use IL-6 blockade and LDN treatment to inhibit hepcidin, to conclude that the latter does not impact the development of Plasmodium in the liver. They have not determined the possibility of IL-6 blockade directly impacting liver-stage malaria, nor the potential timescale of such an impact. No analysis is provided to determine this in Figure 1E, albeit it being just at a single time point.

→ We have determined that neutralizing IL-6 alone did not reverse the suppressive effect that BS infection has on LS infection (see Figure). We didn't include these data in the manuscript due to the fact that neutralizing IL-6 alone did not completely prevent the elevation of hepcidin as shown in Fig. 1B and 2B.

→ As described in our response to the reviewer comment 1, it is not straightforward to assess for how long after the initiation of BS malaria innate cytokine responses might impact LS infection in the mouse malaria models because mice mount protective adaptive immune responses to LS infection. As such we have neutralized the IL-6 during the peak of its expression during BS infection (as per our cytokine kinetic analysis) and yet did not observe a reversal of suppression. This shows that there is no substantial role of BS-induced IL-6 in suppressing LS infection.

4. Antibody-mediated blockade of IFNAR is dose-dependent, as is the case with IFN γ neutralization. With increasing levels of IFN γ or IFNAR-expressing cells, the dose of blocking antibodies required would increase- temporally, further complicated by its local bioavailability. More conclusive and reliable data would be generated using receptor deletion approaches. IFN γ KO, IFNARKO, IFNAR1fl/fl mice, etc., are readily available and some of these strains have been used by the authors in the past. The authors should resort to such precise approaches, especially considering their findings are not in agreement with other previously published data.

→ We agree with the reviewers that neutralization of IFN γ or blockade of IFNAR is antibody dose-dependent. However, we have used high doses of antibodies, everyday throughout the course of the treatment, higher than the effective doses and the timing reported in previous blood stage malaria studies (Sun, Holowka et al. 2012, Sebina, James et al. 2016). Furthermore, we have added IFA analysis of infected liver sections obtained from mice in which IFN γ was neutralized during Py BS infection (Fig.S2). This shows restoration of normal LS development, indicating that the antibody doses we have used were very effective in neutralizing the cytokine.

→ We agree with the reviewer on the importance of using knockout mice as well and have validated the results we obtained with the antibody experiments using IFN γ knockout mice or IFNAR knockout mice, and the data are incorporated in the revised manuscript (Fig.3D and Fig.6A).

5. The key statistics in Fig S1F comparing the 3rd and 4th group are missing. Control wild-type mice for Fig 2D are missing.

→ We have updated the statistics in Fig. S1F, which is now Fig.2F in the revised manuscript.

- We have added the control wild-type mice for Fig.2D which is now Fig.3D in the revised manuscript.
6. Considering the timeline of the experiments, the IFN γ produced by innate immune responses to BS malaria would be expected to impede LS malaria. Nevertheless, this is not a certainty and the extent to which innate and adaptive responses elicited by BS malaria would drive the phenotype observed would expand the scope of the manuscript.
- We agree that it would be interesting to extensively dissect the issue raised by the reviewer. However, to the best of our knowledge, there is a considerable amount of work done already with both rodent malaria and human malaria studies that report the cells types involved in producing IFN γ during *Plasmodium* BS infection (Choudhury, Sheikh et al. 2000, Artavanis-Tsakonas and Riley 2002). More importantly, we are now showing that type I interferons play a significant role along with IFN γ in the suppressive effect that Pb BS infection has on LS infection (Fig.6E, Fig.S4). Type I IFNs are also produced by non-immune cells such as hepatocytes (Sebina and Haque 2018), which further increases the complexity of studies to identify the responsible cell types during BS infection. As such, identifying the cell types that are responsible for producing these cytokines to suppress LS infection is beyond the scope of our study.

Reviewer #2 (Remarks to the Author):

This study shows that 1, Concurrent blood stage (BS) infection inhibits liver stage development; 2, IFN-g is critical in controlling liver stage (LS) development in two rodent malaria parasite species; 3, Heparin has no role in the inhibition of LS development. The observations are interesting and may provide important information for vaccine development. It has been reported that blood-stage infections can hinder the development of protective responses to malaria in a PfSPZ-CVac efficacy study. There are also plenty of reports showing the protective roles of IFN-g in malaria. This study provides data suggesting that the lack of protection in the PfSPZ-CVac efficacy study could be due to increased IFN-g levels induced by BS.

1. The experiments clearly show that IFN-g is critical in controlling LS development. However, it is possible the IFN-g needs to function with other cytokines/chemokines and immune cells to achieve the inhibition of LS. Injection of IFN-g before LS infection may provide some answers to this question.
- We agree as IFN γ is a pleiotropic and highly potent inflammatory cytokine with a range of effects on various immune as well as non-immune cell compartments and it has been shown that injection of IFN γ into mice significantly suppresses LS infection (Nussler, Rénia et al. 1993). To explore the reviewer's question, we have now conducted *in vitro* experiments of the impact of IFN γ on LS infection. We show that it suppresses infection in the absence of other cytokines and without immune cells (Fig.4). Interestingly, our data also show that IFN γ does not affect the early phases of hepatocyte infection such as invasion but instead suppresses LS development (Fig.4). We further

show through immunofluorescence assays (IFA) with infected mouse liver sections that neutralizing IFN γ during Py BS infection restores normal development of LS (Fig.S2).

2. High levels of IFN-g can be induced by infections with other pathogens such as viruses and bacteria. The authors may want to infect mice with a virus or another parasite to induce IFN-g before LS infection to test this possibility.

→ We agree that IFN γ can be induced by other pathogens which might also suppress the LS infection provided the IFN γ responses are generated either systemically or localized to the liver. However, our work investigates the impact of *Plasmodium* BS infection on LS infection and as such, investigating other pathogens is beyond the scope of our study.

3. Another issue is strain-specific immunity that has been demonstrated in malaria parasite infections. If it is only the IFN-g level that affects SPZ vaccine efficacy, strain-specific immunity should not be an issue. It would be nice to infect the host with one strain of *P. yoelii* BS and then infect with LS of another *P. yoelii* strain or *P. berghei*.

→ We have done the experiments using two different rodent malaria parasite species in a heterologous BS/LS coinfection model and demonstrated that BS infection with one species that induces IFN γ suppresses LS development of the same species and a different species. Interestingly, we observed that *Pb* BS infection induces a distinct cytokine responses when compared to *Py* BS infection and further showed that the *Pb* BS-mediated LS suppression is also caused by type-I IFN signaling together with IFN γ . The data is incorporated in the revised manuscript (Fig.5, 6, S4). Questions regarding the acquisition of adaptive immunity and strain specificity of such responses are beyond the scope of our current work.

4. Line 103-107: If the high level of IFN-g can suppress the growth of vaccine sporozoites, then it can also protect the child from natural infection. In this case, why do we need to vaccinate people with BS infection?

→ The scenario in malaria endemic regions is complex and difficult to completely recapitulate with animal models. Mouse models have limitations, and we acknowledge this in the “limitation of the study” section, as part of the discussion in the revised manuscript. We believe that in high transmission areas of malaria, the LS-suppressive effect of blood stage infection is indeed occurring but that it might not be complete. It will be interesting in the future to study this in humans, but the issue is more complex with the acquisition of adaptive immunity.

5. Line 108-111. It would be nice to include experiments after curing active BS infection before liver stage infection.

→ We have performed experiments in which mice infected with *Py* BS infection were treated with chloroquine. LS infection after BS cure was not significantly impacted any longer (see figure). We have not added these data to the revised manuscript because we feel it is not integral to the theme of our current study.

6. As a paper for nature Communications usually requires elucidation of the mechanism, it would be nice to present data showing how IFN- γ inhibits LS growth.

→ It has been shown that IFN γ inhibits the growth of *Plasmodium* LS by inducing nitric oxide production and/or autophagy (Mellouk, Green et al. 1991, Nussler, Rénia et al. 1993, Mellouk, Hoffman et al. 1994, Seguin, Klotz et al. 1994, Lelliott and Coban 2016). A such, we feel that further investigation of this question in the context of our current study is not necessary.

7. Malaria parasite infection also induces immune inhibition including inhibition of many T and B cell pathways. The suppression of the immune system by prior infection may also affect antibody responses even sporozoites can develop normally. The lack of antibody response after vaccination may be due to a general mechanism of immune inhibition. The inhibition can affect vaccine efficacy for all types of vaccines including molecular vaccines without the need for LS development.

→ We completely agree with the reviewer that the impact of malaria BS infection on the generation of vaccine-mediated immunity is much more complex, particularly in humans. In fact, understanding the impact of BS infection on whole sporozoite vaccine mediated adaptive immunity is part of our future plans. Our current study is focused on how concurrent BS infection suppresses LS development via innate mechanisms. This will have direct consequences for the efficacy of replication competent parasite vaccine as it depends on LS growth of the whole cell immunogen (Butler, Schmidt et al. 2011, Sack, Keitany et al. 2015).

Reviewer #3 (Remarks to the Author):

The manuscript by Patel et al describes that infection initiated by *Py* and *Pb* blood stages inhibits subsequent liver infection by sporozoites (~100-fold decrease in parasite load by bioluminescence in *Py* and ~10-fold decrease in *Pb*).

This inhibitory phenotype was first characterized by Portugal et al 2011 and the level of inhibition correlated with the increase in hepcidin levels in *Pb*NK65 infected mice. Over-expression of hepcidin in naïve mice decreased liver infection by ~50% (~2-fold). Based on these results Portugal et al suggested that up-regulation of hepcidin during blood infection inhibited liver infection by sporozoites. Notably lack of IFN γ in knockout mice could not revert this inhibitory phenotype.

On the contrary, in the submitted manuscript, Patel et al show that anti-IFN γ antibodies and

lack of IFN γ in knockout mice can revert the liver infection inhibition phenotype caused by PyXNL blood infection. Additionally, they could not revert liver infection inhibition in blood infected mice by blocking the up regulation of hepcidin using anti-IL6 antibodies + LDN inhibitor. Finally, the authors conclude that “malaria blood stage infection suppresses liver stage infection via IFN γ but not hepcidin”.

This is an important article that challenges the actual model of liver infection inhibition by hepcidin in blood infected mice with potential consequences for live attenuated sporozoite vaccination as well as for better understanding the dynamics of multiple re-infections in the field.

However before recommending this article for publication, I would like to clarify some points.

→ We thank the reviewer for the positive view of our manuscript.

In this manuscript, the authors use a mutant parasite PyGAP^{luc} displaying a late growth impairment to measure “liver infection”. They generalize the conclusion of their findings regarding the effect of IFN γ using this mutant - as can be seen in the title of the manuscript. An experiment using PyXNL^{luc} sporozoite infection should be performed to confirm that the role of IFN γ is not associated with the PyGAP^{luc} phenotype.

→ We thank the reviewer for this suggestion. We have conducted the experiment using wildtype Py^{GFP-luc} or Pb ANKA^{luc} sporozoite infection and obtained very similar results compared to those we had observed with Py GAP^{luc} parasites. The data is included in the revised manuscript (Fig.3E, Fig.5C).

Given the fact that IFN γ KO mice did not revert the phenotype of liver inhibition in Portugal et al 2011, it would be judicious to test anti-IFN γ antibodies and IFN γ KO mice using PbNK65 blood infection and PbANKA^{luc} sporozoite challenge as performed in the supplementary figure1 for anti-IL6/LND inhibitor treatment. This experiment will also provide evidence that the IFN γ role is parasite species transcendent, and it is not only valid for Py.

→ We are glad that the reviewer raises this point. We have now done the experiments in C57BL/6 mice infected with Pb NK65 BS and Pb ANKA^{luc} LS infection and demonstrate that in Pb BS infection, not only IFN γ but also type-I IFN (IFN-I) signaling suppresses LS development (Fig.6E). IFN γ neutralization together with IFNAR1 blockade prevents suppression of LS development during Pb BS infection but either one alone does not. This is a substantial difference between Pb BS and Py BS infection.

It is important to measure the serum concentration of IFN γ in the Py and Pb experiments like the group did in Miller et al, 2014, to verify if they correspond to the concentration required to block liver infection, as well as to confirm the IFN γ -/- phenotype of KO mouse. Equally, parasitemia should be controlled such as in the experiments using anti-IL6/LDN inhibitor.

→ We have now done a comprehensive comparative multi-cytokine kinetic analysis of eleven different analytes, including IFN γ , during the course of Py or Pb BS infection and reveal that the two parasite species provoke distinct host responses (Fig.5D-F).

→ We have added the parasitemia count for the anti-IFN γ neutralization in the revised manuscript and show it has no impact (Fig.3C). It was also previously shown that anti-IFN γ neutralization does not significantly affect blood parasitemia (Sun, Holowka et al. 2012).

In figure 2D, there is almost a 5-fold difference between control and infected IFN γ KO mice, this is almost the difference found in Pb (SFig. 1E). Nonetheless, the figure shows a non-significant difference. Could you please check the statistics? The mean and SD suggest that these values are statistically different.

→ The figure has been updated with proper statistics and controls (Fig.3D).

To investigate the effect of hepcidin on Py sporozoite liver infection, the authors sought to inhibit the signaling pathways responsible for hepcidin induction using anti IL-6 blocking antibodies and LDN193189. They report the effect of the treatment on serum hepcidin level and parasitemia at day 6 post-infection and treatment while Py sporozoite challenge is performed at day 4 post-infection. The authors should show the level of hepcidin at the day of sporozoite challenge as presented for the experiment using Pb.

→ The experiment has been repeated while measuring the hepcidin levels at day 4 post-infection. The figure is updated with new data (Fig. 1B, C).

Minor comments:

The authors convincingly demonstrate that sporozoite infection in mice exhibiting concurrent blood stage infection results in reduced hepatic parasite burden at 40 h post-infection. Analysis at earlier time points is not presented here and an impact of blood stage infection on sporozoite homing to the liver, crossing into the hepatic parenchyma or phagocytosis in the bloodstream therefore cannot be ruled out. Sentences stating that blood stage infection suppresses parasite development (line 54 and line 56) should therefore be reformulated to take into consideration this possibly broader impact on liver stage infection (e.g., by substituting “LS development” by “liver infection”, as in the title of the manuscript).

→ We completely agree with reviewer that we cannot rule out the involvement of the phagocytic immune cells in impacting the process of sporozoites homing to the liver and invading the hepatocytes. It is very challenging to detect the early forms of LS in the liver sections through IFAs, especially when there is concurrent BS parasite infection. To study the impact of IFN γ further however, we have now performed an *in vitro* infection study and determined that the IFN γ does not affect the early phases of hepatocyte infection such as invasion but instead suppresses LS development (Fig.4). Thus, we now use the term ‘suppresses LS infection’ up to the point in the manuscript where we show the *in vitro* data and then use the term ‘suppresses LS development’. We hope this is acceptable to the reviewer.

Could the authors elaborate on why different inoculum of Py and Pb iRBCs were used and why the timings of sporozoite infection were different?

→ The Py BS/Py GAP LS model has originally been established by us with a 10^5 Py iRBCs inoculum. However, in order to follow the exact experimental conditions of Portugal et al 2011 paper we had to use the Pb iRBCs inoculum and the timings of sporozoite infection, which was different. Nevertheless, the inoculum size of 10^5 vs 10^6 did not affect the outcome of the experiment. We have now included data from the experiments that use the same parasite inoculum for both Py and Pb BS and the timing of sporozoite challenge and obtained very similar results. The data are incorporated in the revised manuscript (Fig.5C, Fig.S4).

In the material and methods section, it's unclear which doses of anti-IFN gamma and anti-IFNAR antibodies were administered to the mice to abrogate signaling through these pathways.

→ The details have been incorporated in the revised manuscript.

In Supplementary Fig 1, statistical significance is not indicated on all figures, symbols are missing above some of the bars.

→ The figures have been updated to resolve the issue (Fig.2).

In Supplementary Fig 2B, the mention of which of the two quantifications refers to the group treated with isotype or anti-IFNAR antibodies is missing on the graph.

→ The figures have been updated to resolve the issue (Fig. S1).

References for the above response:

Artavanis-Tsakonas, K. and E. M. Riley (2002). "Innate immune response to malaria: rapid induction of IFN-gamma from human NK cells by live Plasmodium falciparum-infected erythrocytes." J Immunol **169**(6): 2956-2963.

Belnoue, E., T. Voza, F. T. Costa, A. C. Grüner, M. Mauduit, D. S. Rosa, N. Depinay, M. Kayibanda, A. M. Vigário, D. Mazier, G. Snounou, P. Sinnis and L. Rénia (2008). "Vaccination with live Plasmodium yoelii blood stage parasites under chloroquine cover induces cross-stage immunity against malaria liver stage." J Immunol **181**(12): 8552-8558.

Butler, N. S., N. W. Schmidt, A. M. Vaughan, A. S. Aly, S. H. Kappe and J. T. Harty (2011). "Superior antimalarial immunity after vaccination with late liver stage-arresting genetically attenuated parasites." Cell Host Microbe **9**(6): 451-462.

Choudhury, H. R., N. A. Sheikh, G. J. Bancroft, D. R. Katz and J. B. De Souza (2000). "Early nonspecific immune responses and immunity to blood-stage nonlethal Plasmodium yoelii malaria." Infect Immun **68**(11): 6127-6132.

D'Ombra, M. C., L. J. Robinson, D. I. Stanicic, J. Taraika, N. Bernard, P. Michon, I. Mueller and L. Schofield (2008). "Association of early interferon-gamma production with immunity to clinical malaria: a longitudinal study among Papua New Guinean children." Clin Infect Dis **47**(11): 1380-1387.

Deloron, P., C. Chougnet, J. P. Lepers, S. Tallet and P. Coulanges (1991). "Protective value of elevated levels of gamma interferon in serum against exoerythrocytic stages of *Plasmodium falciparum*." J Clin Microbiol **29**(9): 1757-1760.

Lelliott, P. M. and C. Coban (2016). "IFN- γ protects hepatocytes against *Plasmodium vivax* infection via LAP-like degradation of sporozoites." Proc Natl Acad Sci U S A **113**(25): 6813-6815.

Luty, A. J., B. Lell, R. Schmidt-Ott, L. G. Lehman, D. Luckner, B. Greve, P. Matousek, K. Herbich, D. Schmid, F. Migot-Nabias, P. Deloron, R. S. Nussenzweig and P. G. Kremsner (1999). "Interferon-gamma responses are associated with resistance to reinfection with *Plasmodium falciparum* in young African children." J Infect Dis **179**(4): 980-988.

Mellouk, S., S. J. Green, C. A. Nancy and S. L. Hoffman (1991). "IFN-gamma inhibits development of *Plasmodium berghei* exoerythrocytic stages in hepatocytes by an L-arginine-dependent effector mechanism." J Immunol **146**(11): 3971-3976.

Mellouk, S., S. L. Hoffman, Z. Z. Liu, P. de la Vega, T. R. Billiar and A. K. Nussler (1994). "Nitric oxide-mediated antiplasmodial activity in human and murine hepatocytes induced by gamma interferon and the parasite itself: enhancement by exogenous tetrahydrobiopterin." Infect Immun **62**(9): 4043-4046.

Nussler, A. K., L. Rénia, V. Pasquetto, F. Miltgen, H. Matile and D. Mazier (1993). "In vivo induction of the nitric oxide pathway in hepatocytes after injection with irradiated malaria sporozoites, malaria blood parasites or adjuvants." Eur J Immunol **23**(4): 882-887.

Robinson, L. J., M. C. D'Ombra, D. I. Stanicic, J. Taraika, N. Bernard, J. S. Richards, J. G. Beeson, L. Tavul, P. Michon, I. Mueller and L. Schofield (2009). "Cellular tumor necrosis factor, gamma interferon, and interleukin-6 responses as correlates of immunity and risk of clinical *Plasmodium falciparum* malaria in children from Papua New Guinea." Infect Immun **77**(7): 3033-3043.

Sack, B. K., G. J. Keitany, A. M. Vaughan, J. L. Miller, R. Wang and S. H. Kappe (2015). "Mechanisms of stage-transcending protection following immunization of mice with late liver stage-arresting genetically attenuated malaria parasites." PLoS Pathog **11**(5): e1004855.

Sebina, I. and A. Haque (2018). "Effects of type I interferons in malaria." Immunology **155**(2): 176-185.

Sebina, I., K. R. James, M. S. Soon, L. G. Fogg, S. E. Best, F. Labastida Rivera, M. Montes de Oca, F. H. Amante, B. S. Thomas, L. Beattie, F. Souza-Fonseca-Guimaraes, M. J. Smyth, P. J. Hertzog, G. R. Hill, A. Hutloff, C. R. Engwerda and A. Haque (2016). "IFNAR1-Signalling Obstructs ICOS-mediated Humoral Immunity during Non-lethal Blood-Stage *Plasmodium* Infection." PLoS Pathog **12**(11): e1005999.

Seguin, M. C., F. W. Klotz, I. Schneider, J. P. Weir, M. Goodbary, M. Slayter, J. J. Raney, J. U. Aniagolu and S. J. Green (1994). "Induction of nitric oxide synthase protects against malaria in mice exposed to irradiated *Plasmodium berghei* infected mosquitoes: involvement of interferon gamma and CD8+ T cells." J Exp Med **180**(1): 353-358.

Senkpeil, L., J. Bhardwaj, M. Little, P. Holla, A. Upadhye, P. A. Swanson, R. E. Wiegand, M. D. Macklin, K. Bi, B. J. Flynn, A. Yamamoto, E. L. Gaskin, D. N. Sather, A. L. Oblak, E. Simpson, H. Gao, W. N. Haining, K. B. Yates, X. Liu, K. Otieno, S. Kariuki, X. Xuei, Y. Liu, R. Polidoro, S. L. Hoffman, M. Onoko, L. C. Steinhardt, N. W. Schmidt, R. A. Seder and T. M. Tran (2021). "Innate immune activation restricts priming and protective efficacy of the radiation-attenuated PfSPZ malaria vaccine." medRxiv: 2021.2010.2008.21264577.

Sun, T., T. Holowka, Y. Song, S. Zierow, L. Leng, Y. Chen, H. Xiong, J. Griffith, M. Nouraie, P. E. Thuma, E. Lolis, C. J. Janse, V. R. Gordeuk, K. Augustijn and R. Bucala (2012). "A *Plasmodium*-encoded cytokine suppresses T-cell immunity during malaria." Proc Natl Acad Sci U S A **109**(31): E2117-2126.

Reviewers' Comments:

Reviewer #1:

Remarks to the Author:

In this revised manuscript also, the key concern is the scope of this manuscript.

The authors emphasize in the manuscript and the response to the reviews that 'the' primary contribution of this work is demonstrating how hepcidin regulation may not be the key pathway through which existing blood-stage infection might impact a new liver-stage Plasmodium infection. This reviewer does not believe that this discovery contradicting a previous finding alone has the broad impact on the field that would warrant consideration in Nature Communications. In this reviewer's view, the discovery that would have elevated it to that level would be how the results may have the potential to alter malaria vaccine strategies.

In the field, where subclinical malaria is common, many people may remain parasitemic. A key contribution of this work according to the authors, which this reviewer wholeheartedly agrees with- is in the context of such people getting vaccinated with live-attenuated Plasmodium strains (e.g. lines 292-296). Understanding the mechanism by which parasitemic individuals generate compromised protective responses following vaccination would allow us to reverse such defects in the field.

A blood stage infection can last a reasonably long time in humans. Often more than a week when untreated. A prolonged course of blood-stage malaria is somewhat true in mice as well, depending on the species of the parasite. This study indicates that the development of Plasmodium in the liver can be negatively impacted by an ongoing blood-stage infection. However, the study is limited to looking at the impact of BS malaria on LS malaria for only up to 1 week. This is an extremely small window and narrow in scope. My main concern is that these data have limited relevance, even if it uncovered the mechanism at play.

Expanding their timeline to later time points would enhance the scope of this work, which, in my view, would make it suitable for publication in a journal with a broad readership and interest such as Nature Communications. The authors responded to this suggestion in the original review that 'since adaptive immune responses that are primed in response to the BS infection will start to play a major role in suppressing the LS infection at later time points (Belnoue, Voza et al. 2008)', the data they obtain may be hard to interpret. However, the cited manuscript indicates that the adaptive responses generated in this manner also act through IFN γ . It is also possible to specifically prevent the development of such adaptive responses to avoid it being a confounding factor.

In Short, this reviewer feels that it is necessary to expand the scope of this work to include later time points (say, up to when parasitemia is not detected anymore, or cleared using drugs).

Other major concerns:

In the new data presented in Fig 6A, there seems to be no difference between the liver parasite burdens between WT, anti-IFNAR1 treated and IFNAR1KO mice. From my reading, this is quite surprising and in stark contrast to their previous published findings from this group (Miller et al). A similar piece of data is presented in Fig 6C as well. The authors would need to discuss this, explaining how/why their new findings appear to contrast their own and others' (Liehl et al, 2014) previous work.

Minor concern:

Please consider using 'parasitemia' instead of 'blood parasitemia' throughout the manuscript. I believe the '-emia' suffix automatically indicates that you are looking within the blood.

Reviewer #2:

Remarks to the Author:

The revision with additional data makes this study more interesting and significant. The data are well-presented.

Minor points/comments.

--To make it easier for readers, please label the parasite and mouse strains used in each figure (mouse infection images).

--Figure 5D. Type I interferons can be induced early (18-24 hpi) or at a later stage (day3-5). Measuring these cytokines at 3-day intervals may miss their peaks of expression. Also, type I interferons include many subspecies. To make it more specific, the authors can consider measuring type I interferons at more time points and more subtypes later. It would not change the major conclusions without the data.

--Figure 4A. Please indicate the amounts of IFN γ used. What are the images in Figure 4C? Similarly, D and F are images of...

--Figure 5B and 5C. The labels for the scale bars are too small.

--Figure S2, why no image for isotype/BS+LSluc?

--Ref 16, 58 missing page numbers

Reviewer #3:

Remarks to the Author:

The authors satisfactorily addressed all the points previously raised which resulted in a substantial amelioration of the manuscript. I thus have no major objections for the experimental part of the first manuscript version.

I just have one major remark and a few minor comments regarding the new manuscript version.

Increased serum levels of IFNs can be transiently observed around 2-6 days post-infection with PbBS (IFN γ and IFN α), or around 1-5 days post-infection with PyBS (IFN γ). These data indicate that the inhibition of LS infection in BS infected mice, could be a transient phenomenon.

Since in the model, sporozoites are injected at D4 post-BS infection, the effect of IFNs is close to its maximum. It would be important to test the LS inhibition (at least in the Py model which depends only on IFN γ) using a sporozoite infection after day 7 post-BS inoculation when levels of IFN γ are back to normal. This would precise if this inhibition can be sustained during BS infection with normal levels of IFN γ or if it is only a transitory phenomenon occurring when IFN γ levels are high during BS infection. This experiment is key to better understand how long one should administer live attenuated sporozoite vaccines following clearance of BS infection and which parameter should be considered to evaluate hepatocyte refractoriness. Of note, in vitro refractoriness of hepatocyte to sporozoite infection can last up to 3 weeks following injection of irradiated sporozoites (Ref 31). Confounding effects cited in lines 313-320 could be addressed by depletion of CD8+ cells using antibodies prior to sporozoite infection or be attenuated by using Py BS and Pb sporozoites (or vice versa). Alternatively and ideally, hepatocyte refractoriness could be evaluated as in Ref 31.

There are some passages where key references are omitted.

General: The seminal work of Ferreira et al, Science 1986 about in vivo LS development and IFN γ is absent in the manuscript.

Line 153: "Unexpectedly, we did not observe restoration of Pb LS development with IFN γ neutralization in the Pb BS + Pb LS group (Fig. 5B).". The formulation is not precise since this result was expected and already published by Portugal et al using IFN γ KO mice.

Line 215: "Interestingly, it was previously shown that hepatocytes harvested from BS infected mice failed to support normal LS infection, suggesting that the BS-induced host responses condition

hepatocytes to become refractory to LS development 31.” It would be important to discuss that Ref 31 also showed that in vivo injection of IFNg causes the refractoriness of hepatocytes to sporozoite infection in vitro, and correct the sentence in line 309: “We have also shown that IFNg renders hepatocytes refractory to support LS development.”

Response to Reviewers

We like to thank reviewers for their positive comments and final constructive critiques. Below, please find the point-by-point responses to each reviewers' critique.

Reviewer #1:

In this revised manuscript also, the key concern is the scope of this manuscript.

The authors emphasize in the manuscript and the response to the reviews that 'the' primary contribution of this work is demonstrating how hepcidin regulation may not be the key pathway through which existing blood-stage infection might impact a new liver-stage Plasmodium infection. This reviewer does not believe that this discovery contradicting a previous finding alone has the broad impact on the field that would warrant consideration in Nature Communications. In this reviewer's view, the discovery that would have elevated it to that level would be how the results may have the potential to alter malaria vaccine strategies.

→ The finding by Portugal et al. that blood stage-induced hepcidin suppresses liver stage development was considered a seminal finding and was published in Nature Medicine. Our findings clearly contradict these findings. As such, this alone constitutes broad impact for our work. However, we agree with the reviewer that our work is much further elevated in impact by uncovering the actual mechanism of suppression, which will inform malaria vaccination strategies.

In the field, where subclinical malaria is common, many people may remain parasitemic. A key contribution of this work according to the authors, which this reviewer wholeheartedly agrees with is in the context of such people getting vaccinated with live-attenuated Plasmodium strains (e.g. lines 292-296). Understanding the mechanism by which parasitemic individuals generate compromised protective responses following vaccination would allow us to reverse such defects in the field.

→ We thank the reviewer and are in full agreement.

A blood stage infection can last a reasonably long time in humans. Often more than a week when untreated. A prolonged course of blood-stage malaria is somewhat true in mice as well, depending on the species of the parasite. This study indicates that the development of Plasmodium in the liver can be negatively impacted by an ongoing blood-stage infection. However, the study is limited to looking at the impact of BS malaria on LS malaria for only up to 1 week. This is an extremely small window and narrow in scope. My main concern is that these data have limited relevance, even if it uncovered the mechanism at play.

Expanding their timeline to later time points would enhance the scope of this work, which, in my view, would make it suitable for publication in a journal with a broad readership and interest such as Nature Communications. The authors responded to this suggestion in the original review that 'since adaptive immune responses that are primed in response to the BS infection will start to play a major

role in suppressing the LS infection at later time points (Belnoue, Voza et al. 2008), the data they obtain may be hard to interpret. However, the cited manuscript indicates that the adaptive responses generated in this manner also act through IFNg. It is also possible to specifically prevent the development of such adaptive responses to avoid it being a confounding factor.

In Short, this reviewer feels that it is necessary to expand the scope of this work to include later time points (say, up to when parasitemia is not detected anymore, or cleared using drugs).

→ We thank the reviewer for this comment. Clinical blood stage infection is common in high malaria transmission areas, and it is very well described that subclinical infections are even more common. As such, our findings have significant relevance. We agree that it is interesting to investigate if the liver stages would develop efficiently when the IFNg responses return back to the normal and/or the parasitemia is not detected. However, it will require substantial and protracted investigation and as such, is beyond the scope of this manuscript. We do show that IFNg renders hepatocytes refractory to the subsequent sporozoite infection as (Fig. 4A, D-G, 16h Pre-treatment group), which was also observed in a previous study (Nussler, Rénia et al. 1993). This suggests that a liver stage suppressive effect of IFNg might persist when the cytokine levels return to normal and sets the stage for interesting future studies.

Other major concerns:

In the new data presented in Fig 6A, there seems to be no difference between the liver parasite burdens between WT, anti-IFNAR1 treated and IFNAR1KO mice. From my reading, this is quite surprising and in stark contrast to their previous published findings from this group (Miller et al). A similar piece of data is presented in Fig 6C as well. The authors would need to discuss this, explaining how/why their new findings appear to contrast their own and others' (Liehl et al, 2014) previous work.

→ We show indeed in Figure 6A that IFN-I signaling alone is not sufficient to explain the liver stage-suppressive effect of a Pb blood stage infection. We then show that during Pb blood stage infection, it is a combination of IFN-I signaling AND IFN-g, which together suppress liver stage infection. The infection model used here is very different from that used in our Miller et al. work. In the latter we used two consecutive liver stage infections and showed that IFN-I signaling induced by the first liver stage infection, suppresses the second liver stage infection. This occurs via NKT cells that secrete IFNg. We are not certain where the reviewer sees the big contrast, particularly since the assumption that blood stage versus liver stage-induced innate inflammation would be the same in terms of the diversity and the magnitude of the responses, is an oversimplification

Minor concern:

Please consider using 'parasitemia' instead of 'blood parasitemia' throughout the manuscript. I believe the '-emia' suffix automatically indicates that you are looking within the blood.

→ We have addressed the issue in the revised manuscript.

Reviewer #2 (Remarks to the Author):

The revision with additional data makes this study more interesting and significant. The data are well-presented.

→ We thank the reviewer for the positive view of our manuscript.

Minor points/comments.

To make it easier for readers, please label the parasite and mouse strains used in each figure (mouse infection images).

→ The figures have been updated to address this comment.

Figure 5D. Type I interferons can be induced early (18-24 hpi) or at a later stage (day3-5). Measuring these cytokines at 3-day intervals may miss their peaks of expression. Also, type I interferons include many subspecies. To make it more specific, the authors can consider measuring type I interferons at more time points and more subtypes later. It would not change the major conclusions without the data.

→ We had carefully chosen the timepoints to measure selected cytokines during the course of Py and Pb blood stage infection. The data in Fig 5D represent day 0, 1, 3, 4, 6, 9, 15, and 18 of the blood stage infection. We have also attached the source data file to give readers precise expression values for each analyte shown in the figure. We have measured IFN α and IFN β cytokines during the course of blood stage infection. We did not observe significant expression of IFN β (data included in the Fig 5 tab of the source data Excel file), whereas we observed high levels of IFN α only in Pb BS infected mice. Dissecting out the subtype of IFN α is beyond the scope of the current manuscript.

Figure 4A. Please indicate the amounts of IFN γ used. What are the images in Figure 4C? Similarly, D and F are images of...

→ We have addressed the issue in the revised manuscript.

Figure 5B and 5C. The labels for the scale bars are too small.

→ We have addressed the issue in the revised manuscript.

Figure S2, why no image for isotype/BS+LSluc?

→ We couldn't visually detect any parasite in isotype/BS+LS^{luc} group with 50,000 wildtype *Py*^{GFP-luc} sporozoites inoculum. We had noted 'ND': 'not detected' in the figure and its legend.

Ref 16, 58 missing page numbers

→ Thank you, for noticing. We have updated the reference with page numbers.

Reviewer #3 (Remarks to the Author):

The authors satisfactorily addressed all the points previously raised which resulted in a substantial amelioration of the manuscript. I thus have no major objections for the experimental part of the first manuscript version.

→ We thank the reviewer for the positive view of our manuscript.

I just have one major remark and a few minor comments regarding the new manuscript version.

Increased serum levels of IFNs can be transiently observed around 2-6 days post-infection with PbBS (IFN γ and IFN α), or around 1-5 days post-infection with PyBS (IFN γ). These data indicate that the inhibition of LS infection in BS infected mice, could be a transient phenomenon.

Since in the model, sporozoites are injected at D4 post-BS infection, the effect of IFNs is close to its maximum. It would be important to test the LS inhibition (at least in the Py model which depends only on IFN γ) using a sporozoite infection after day 7 post-BS inoculation when levels of IFN γ are back to normal. This would precise if this inhibition can be sustained during BS infection with normal levels of IFN γ or if it is only a transitory phenomenon occurring when IFN γ levels are high during BS infection. This experiment is key to better understand how long one should administer live attenuated sporozoite vaccines following clearance of BS infection and which parameter should be considered to evaluate hepatocyte refractoriness. Of note, in vitro refractoriness of hepatocyte to sporozoite infection can last up to 3 weeks following injection of irradiated sporozoites (Ref 31). Confounding effects cited in lines 313-320 could be addressed by depletion of CD8 $^{+}$ cells using antibodies prior to sporozoite infection or be attenuated by using Py BS and Pb sporozoites (or vice versa). Alternatively, and ideally, hepatocyte refractoriness could be evaluated as in Ref 31.

→ We appreciate the reviewers' inquiry. It is possible that the effect of blood stage induced IFNs could be transient, or the effect might persist for longer period of time in the form of refractoriness of the hepatocytes to the subsequent sporozoite infection as we observed in Fig. 4A, D-G (16h Pre-treatment group) and according to the previous study (Nussler, Rénia et al. 1993). Clinical blood stage infection is common in high malaria transmission areas, and it is very well described that subclinical infections are even more common. As such, our current findings have significant relevance. We agree that it is interesting to investigate if the liver stages would develop efficiently when the IFN γ responses return back to the normal and/or the parasitemia is not detected. However, it will require substantial and protracted investigation and as such, is beyond the scope of this manuscript.

There are some passages where key references are omitted.

General: The seminal work of Ferreira et al, Science 1986 about in vivo LS development and IFN γ is absent in the manuscript.

→ The work has now been cited in the revised manuscript (Ref#20).

Line 153: “Unexpectedly, we did not observe restoration of Pb LS development with IFN γ neutralization in the Pb BS + Pb LS group (Fig. 5B).”. The formulation is not precise since this result was expected and already published by Portugal et al using IFN γ KO mice.

→ The issue has been resolved in the revised manuscript.

Line 215: “Interestingly, it was previously shown that hepatocytes harvested from BS infected mice failed to support normal LS infection, suggesting that the BS-induced host responses condition hepatocytes to become refractory to LS development 31.” It would be important to discuss that Ref 31 also showed that in vivo injection of IFN γ causes the refractoriness of hepatocytes to sporozoite infection in vitro, and correct the sentence in line 309: “We have also shown that IFN γ renders hepatocytes refractory to support LS development.”

→ We have addressed the comment in the revised manuscript (Line 251 – 255).